# Towards sustainable solutions: Effective waste classification framework via enhanced deep convolutional neural networks

**Md. Minhazul Islam**[1], **S. M. Mahedy Hasan**[2], **Md. Rakib Hossain**[1], **Md. Palash Uddin** [3]*, **Md. Al Mamun**[2]

1 Department of Electronics and Telecommunication Engineering, Rajshahi University of Engineering & Technology, Rajshahi, Bangladesh, 2 Department of Computer Science and Engineering, Rajshahi University of Engineering & Technology, Rajshahi, Bangladesh, 3 Department of Computer Science and Engineering, Hajee Mohammad Danesh Science and Technology University, Dinajpur, Bangladesh

* palash_cse@hstu.ac.bd

**Data availability statement:** The datasets used in this work are publicly accessible at

## Abstract

As industrialization and the development of smart cities progress, effective waste collection, classification, and management have become increasingly vital. Recycling processes depend on accurately identifying and restoring waste materials to their original states, essential for reducing pollution and promoting environmental sustainability. In recent years, deep learning (DL) techniques have been applied strategically to enhance waste management processes, including capturing, classifying, composting, and disposing of waste. In light of the current context, the study presents an innovative waste classification model that utilizes a tailored DenseNet201 architecture coupled with an integrated Squeeze and Excitation (SE) attention mechanism and the fusion of parallel Convolutional Neural Network (CNN) branches. The integration of SE attention enables squeezing the irrelevant features and excites the important ones and the fusion of parallel CNN branches enhances the extraction of intricate, deeper, and more distinguishable features from waste data. The evaluation of the model across four publicly available datasets, along with three additional datasets to enhance waste diversity and the model's reliability, and the incorporation of Grad-CAM to visualize and interpret the model's focus areas for transparent decision-making, confirms its effectiveness in improving waste management practices. Furthermore, this model's successful deployment in a web-based sorting system marks a tangible stride in translating theoretical advancements into on-the-ground implementation, promising heightened efficiency and scalability in waste management practices. This work presents a precise solution for adaptable waste classification, heralding a paradigm shift in global waste disposal norms.

https://www.kaggle.com/ds/2814684, https://www.kaggle.com/datasets/mostafaabla/ garbage-classification, https://github.com/openrecycle/dataset, https://github.com/garythung/trashnet, https://www.kaggle.com/datasets/angelikasita/ waste-images, https://www.kaggle.com/ datasets/sanjadrag24/recyclable-waste-images, and https://github.com/sam-single/realwaste.

**Funding:** The author(s) received no specific funding for this work.

**Competing interests:** Authors have no conflict of interest to declare.

## 1 Introduction

Effective waste management poses a significant global challenge, as improper handling and disposal raise environmental, economic, and societal concerns. As emphasized by the World Bank [1], solid waste management is a substantial concern for cities worldwide, with far-reaching consequences. Inadequate waste management not only leads to the emission of greenhouse gases, soil contamination, and depletion of crucial natural resources, threatening the environment, but also results in economic repercussions such as increased healthcare expenses, reduced tourism revenue, and diminished property values. A 2017 World Bank report estimated the global cost of inadequate waste management to be between 222 billion and 370 billion dollars annually [2], encompassing expenses related to waste collection, transportation, disposal, and environmental damage. The gravity of the issue extends beyond financial burdens, impacting communities and ecosystems.

Future projections are alarming, with waste management costs expected to rise to 400 billion dollars per year by 2025. Moreover, inadequate waste management can exacerbate societal inequalities, facilitate the spread of disease vectors, and disrupt local communities. The escalating generation of waste, driven by rapid urbanization and population growth, underscores the urgency of implementing effective waste classification and sustainable waste management practices [3]. Addressing these challenges is crucial for safeguarding the environment and ensuring communities' well-being and economies' resilience [4–6]. Similar proactive measures are increasingly being adopted across various fields to enhance overall efficiency [7].

Traditional methods of waste classification have traditionally relied on manual sorting and visual inspection as the primary means of categorizing waste materials. While visual inspection is widely practiced, it suffers from subjectivity and is labor-intensive, often resulting in errors and inconsistencies in the classification of waste items. On the other hand, laboratory testing, although more accurate, is considerably more expensive and time-consuming. A comprehensive study conducted by Kang et al. [8] emphasizes the pressing need for more accurate and efficient waste classification methods. As highlighted in the study, traditional approaches often fall short of providing the precision required in modern waste management systems. This underscores the imperative for the development and adoption of innovative and practical waste classification techniques that can overcome the shortcomings of traditional methods and meet the demands of contemporary waste management practices.

Researchers have been actively involved in the development of computer-aided decision support systems (DSSs) that leverage the capabilities of artificial intelligence (AI), deep learning (DL), and machine learning (ML) for waste classification. In the field of waste management, there is a growing adoption of advanced DL techniques, including convolutional neural networks (CNNs) and deep belief networks (DBNs), as exemplified by studies conducted by the authors of [9,10]. Previous research efforts in waste classification have primarily relied on conventional ML techniques [11,12]. These traditional methods have been instrumental in tasks like image segmentation, feature extraction, and pattern recognition, but they often face challenges in handling large and diverse datasets effectively.

Deep learning (DL) models, characterized by their deep-layer architectures, excel in abstracting data efficiently, especially when dealing with image datasets. As these developing AI systems have access to large annotated image datasets for efficient training, they demonstrate the potential to improve waste classification accuracy and efficiency. The accuracy of these systems is highly dependent on various factors, including the diversity of

waste elements in training data and the quality of images. Although deep learning (DL) techniques are widely used for waste categorization, there is a noticeable lack of research on the parallel CNN strategy that uses squeezing and excitation. This method offers the potential to outperform traditional DL methods in terms of quality and optimization. Additionally, the majority of existing research in this field does not take into account the deployment of DL approaches for applications in real life, which is essential for evaluating their scalability and practical efficacy. Addressing these limitations could provide valuable insights for advancing waste classification systems using deep learning methodologies.

In this context, this paper presents a comprehensive evaluation of four distinct datasets for waste classification, along with three additional datasets to enhance diversity and reliability, utilizing multiple state-of-the-art architectures. Our research aims to address the fundamental challenge of accurately and efficiently classifying waste materials by harnessing the power of modern DL techniques. Through rigorous experimentation and analysis, we have achieved noteworthy outcomes that promise to enhance the effectiveness of waste classification systems. The primary contributions of this study can be summarized as follows:

- The research utilizes DenseNet201, combined with a parallel CNN architecture and a squeeze-and-excitation attention mechanism, for waste classification. The parallel approach enhances feature extraction by processing data through multiple pathways concurrently, while the squeeze-and-excitation mechanism dynamically refines feature importance, optimizing classification accuracy and efficiency.
- The optimized parallel approach model sets a new standard in waste classification research through its evaluation across four diverse datasets, supplemented by three additional datasets to further enhance diversity, diverging from conventional single-dataset assessments. This multi-dataset approach enhances the model's reliability and applicability across various waste compositions and scenarios, bolstering confidence in its real-world deployment.
- Grad-CAM visualization has been incorporated to interpret the model's decision-making process by highlighting the discriminative regions in input images, thereby enhancing the explainability and trustworthiness of the classification results.
- A user-friendly web system has been developed as a practical tool for waste classification, showcasing its real-world applicability. Currently hosted on a local server, the website demonstrates the potential for broader deployment to assist in waste management endeavors, thereby contributing to more effective and efficient waste management practices.
- In comparison to conventional website development methodologies, Gradio presents a notably streamlined approach for real-time waste classification. Its streamlined deployment procedure makes it the best choice for fast, ad hoc analysis, highlighting its potential for extensive integration into waste management programs.

The remaining part of the article is organized as follows. The "Related work" section reviews prior research, while the "Dataset" section details the datasets used. The "Research methodology" section provides a detailed explanation of the approach and techniques used in this study, followed by the "Result analysis" section, which presents and discusses experimental results. The "Real-time deployment of waste classification system" and "Threats to validity" sections analyze the real-time implementation and potential research limitations. Finally, the "Conclusion and future research" section summarizes key findings and suggests future directions.

## 2 Related work

Waste classification, as a subfield of environmental science and ML, has witnessed significant attention from researchers due to its critical importance in addressing waste management challenges. This section provides an overview of relevant literature, highlighting key studies, methodologies, and technological advancements that have paved the way for our research.

In Meng et al.'s study [10], the investigation of waste image classification encompassed several ML models, such as a Support Vector Machine (SVM), a basic CNN, ResNet50, a simplified ResNet50 network, and a hybrid model combining a histogram of gradient (HOG) features with CNN. Utilizing a dataset consisting of 2,527 images spanning six recycling categories, ResNet50 emerged as the top performer, achieving a notable test accuracy of 95.35%, surpassing the performance of the simple CNN and straightforward network configurations. However, the study faced significant limitations, including a relatively small dataset size, raising concerns about potential overfitting and the ability of the models to accurately represent real-world challenges. Additionally, despite the high accuracy achieved by ResNet50, there remains room for improvement in overall accuracy. It is essential to conduct further testing on diverse datasets to ensure the validity and generalizability of the model's performance. Moreover, no Grad-CAM or similar visualization techniques were incorporated, limiting model interpretability.

In their work, the authors of [13] introduced a waste classification system utilizing DL techniques, specifically leveraging the VGG16 model. This system demonstrated the capability to classify domestic waste with a recognition accuracy of 75.6%. While the achieved performance showed promise for practical applications, it falls short of surpassing the performance levels observed in certain other DL projects focused on image recognition and classification. There is ample opportunity for improvement, indicating a need for further refinement to enhance the system's accuracy.

Rabano et al. [14] introduced an image classification model based on MobileNet for sorting various types of common trash, achieving a final test accuracy of 87.2%. However, a key limitation of the study lies in the relatively modest accuracy attained, which may render the model less suitable for practical waste-sorting applications. Thus, there's a clear imperative for further refinement and fine-tuning of the model to bolster its accuracy and thereby enhance its reliability in real-world scenarios; additionally, Grad-CAM can be incorporated to improve interpretability and provide visual insights into the model's decision-making process.

Guo et al. [15] introduced a waste classification framework leveraging the EfficientNet model, employing the Huawei Artificial Intelligence Competition dataset. To enhance recognition accuracy, they incorporated an attention mechanism aimed at highlighting crucial information while reducing the influence of irrelevant details in the images. This strategy yielded a notable average accuracy rate of 93.47%, with the highest recorded accuracy reaching 98.3%. It's noteworthy to mention that despite these promising outcomes, there hasn't been any practical deployment or application of the model at this stage.

In a subsequent study conducted by the authors of [16], various TL models including VGG16, ResNet-50, and Xception were assessed for waste classification employing the TrashNet dataset. The study reported the highest accuracy of 88% with the Xception model, demonstrating superior precision and recall. However, the scarcity of data particularly for the "trash" category presented a notable challenge, affecting the overall accuracy of the models. The findings suggest a need for further refinement of model parameters to enhance their suitability for practical waste management applications. Additionally, evaluating the model's performance using alternative datasets is deemed necessary to validate its robustness and generalizability.

In a separate study referenced as [17], the GAF_dense model demonstrated remarkable performance, achieving an accuracy of 92.1%, surpassing both traditional and lightweight image classification models [18]. This success was attributed to fine-tuning techniques, and the model exhibited robust generalization when tested on CIFAR-10 and ImageNet datasets, thereby broadening its potential applications. However, it's noteworthy that during additional testing, a significant decrease in accuracy was observed, highlighting the necessity for further investigation and refinement. Furthermore, the study did not explore practical applications of the model and didn't incorporated Grad-CAM or any kind to improve interpretability.

Another waste classification and recognition system proposed in [19], based on TL from the Inception-V3 model, achieved notable results with a training accuracy of 99.3% and a test accuracy of 93.2%. However, the study's limitation lies in the discrepancy between the training and test accuracies, indicating potential issues with generalization. This suggests the need for further exploration and refinement to ensure consistent performance across various datasets and real-world scenarios.

The study referenced as [20] aimed to enhance waste classification accuracy by introducing improvements to the VGG16 model, incorporating a novel activation function and MRMR feature selection technique. Notably, the enhanced VGG16 model outperformed the traditional VGG16 in classifying various types of waste, including recyclable waste, hazardous waste, food waste, and residual waste, achieving accuracy percentages ranging from 94.2% to 96.5%. In another work by Shi et al. [21], a multilayer hybrid CNN model was employed for waste classification across six categories. The model achieved a commendable classification accuracy of up to 92.6%. While this is a notable accomplishment, there remains room for improvement in accuracy, suggesting potential avenues for further optimization and refinement of the model.

In another research study [22], researchers enhance waste image classification using the lightweight MobileNetV2 model. The research incorporates preprocessing steps like shearing and zooming, comparing the base model with Softmax to variations employing SVM, global average pooling, and an additional fully connected layer. Results reveal that MobileNetV2 with SVM achieves 94.28% accuracy, while the global average pooling layer enhances generalization to 81.10%. Introducing an extra fully connected layer further improves accuracy to 83.46%. The study reinforces the importance of lightweight models for efficient waste image classification.

Yu et al. [23] led the development of a waste classification algorithm rooted in DL, achieving an impressive 95% accuracy rate. This achievement was attributed to meticulous data enhancement and learning rate optimization. While this accuracy offers promise for waste management applications, it is crucial to acknowledge the potential impact of real-world challenges, such as diverse waste types and fluctuating environmental conditions, on the model's performance. Consequently, further research is indispensable to evaluate the algorithm's robustness in practical waste management scenarios and incorporated visualization techniques to improve interpretability.

In a recent study by Zhao et al. [24] in 2022, an enhanced version of MobileNetV3 combined with LSTM yielded an impressive classification accuracy of 81%. While this achievement is noteworthy, it emphasizes the ongoing pursuit of further enhancement in classification performance. Similarly, Majchrowska et al. [25] explored the potential of the EfficientNet-B2 model, achieving a classification accuracy of 75%. Despite this accomplishment, there remains ample room for improvement, particularly considering that the current model employs a semi-supervised approach, adding complexity to its design.

In another research effort [26], DL models, specifically Densenet121 and Densenet169, were employed for the classification of recyclable waste, achieving impressive test accuracies of 95%. While promising, the study is constrained by a small dataset and lacks real-world testing. Moreover, the compatibility of the model with other datasets remains unexplored, and no real-world applications have been implemented, warranting further research for a comprehensive assessment.

Furthermore, a study [27] introduced a DL system that achieved a remarkable 92.5% accuracy in automating waste classification. The system encompasses data collection, pre-processing, feature extraction, and classification utilizing techniques such as SVM, Random Forests, and Neural Networks. While holding promise for improving waste management and recycling, the study underscores the need for a more extensive waste classification dataset. Future enhancements may involve incorporating advanced deep learning techniques such as reinforcement learning or attention mechanisms to further boost accuracy and efficacy in classification.

The existing literature highlights several research gaps and limitations in the field of waste classification using deep learning models. A significant number of studies are limited by small dataset sizes, inadequate generalization to various real-world contexts, and a lack of thorough investigation into model robustness and practical implementation. Additionally, many studies lack the incorporation of visualization techniques, such as Grad-CAM, which could enhance model interpretability and provide valuable insights into the decision-making process. Common challenges include overfitting, inconsistencies between training and testing accuracies, and a deficiency in practical application or extensive model evaluation. While certain models exhibit high accuracy, they frequently lack the necessary generalizability and applicability in real-world situations. This research seeks to address these deficiencies by utilizing a more extensive and varied dataset, integrating advanced methodologies to enhance model robustness and accuracy, and investigating practical applications to improve performance in real-world settings. These initiatives aim to deliver a more dependable and effective solution for waste classification that is better equipped to tackle the challenges encountered in practical applications.

## 3 Dataset

The study incorporated a diverse range of datasets to ensure comprehensive coverage of waste classification scenarios, with four distinct datasets serving unique purposes in enriching the research's data diversity. Two datasets were sourced from Kaggle [28,29] each featuring waste classification tasks with varying complexities. The first dataset, titled "waste classification V2," also referred as the "Garbage Dataset" on Kaggle, comprised 10 classes and offered a substantial repository of 21,983 images. The second dataset, named "waste classification", also can be called as "Garbage classification (12 classes)" on Kaggle, was more intricate with 12 classes, and contained a rich collection of 15,515 images. In addition to the Kaggle datasets, this research also leveraged the "OpenRecycle" dataset [30], which consisted of 7 classes and further expanded the study's scope with 2,301 images, providing a distinct set of challenges and data distribution. Furthermore, the study included the "TrashNet" dataset [31], which featured 6 classes and a substantial corpus of 2527 images.

To validate the model's reliability and potential while increasing waste diversity, three additional datasets were also considered. The first dataset, from the study by Ouedraogo et al. [32], contains 8,346 images across 9 classes and is publicly available on Kaggle [33], titled "Waste Images." The second dataset, also available on Kaggle [34], is named "Recyclable Waste Images" and consists of 3,180 images across 9 classes. The third dataset, titled "RealWaste,"

sourced from the study by Single et al. [35], includes 4,753 images across 9 classes and is publicly available on GitHub [36]. Fig 1 offers a visual display of sample images from the datasets, while Table 1 provides a concise overview detailing the number of samples and classes for each dataset.

Table 2 presents the unique waste classes identified across the seven datasets, totaling 27 distinct categories. Although the original datasets encompassed 62 classes, these 27 have been

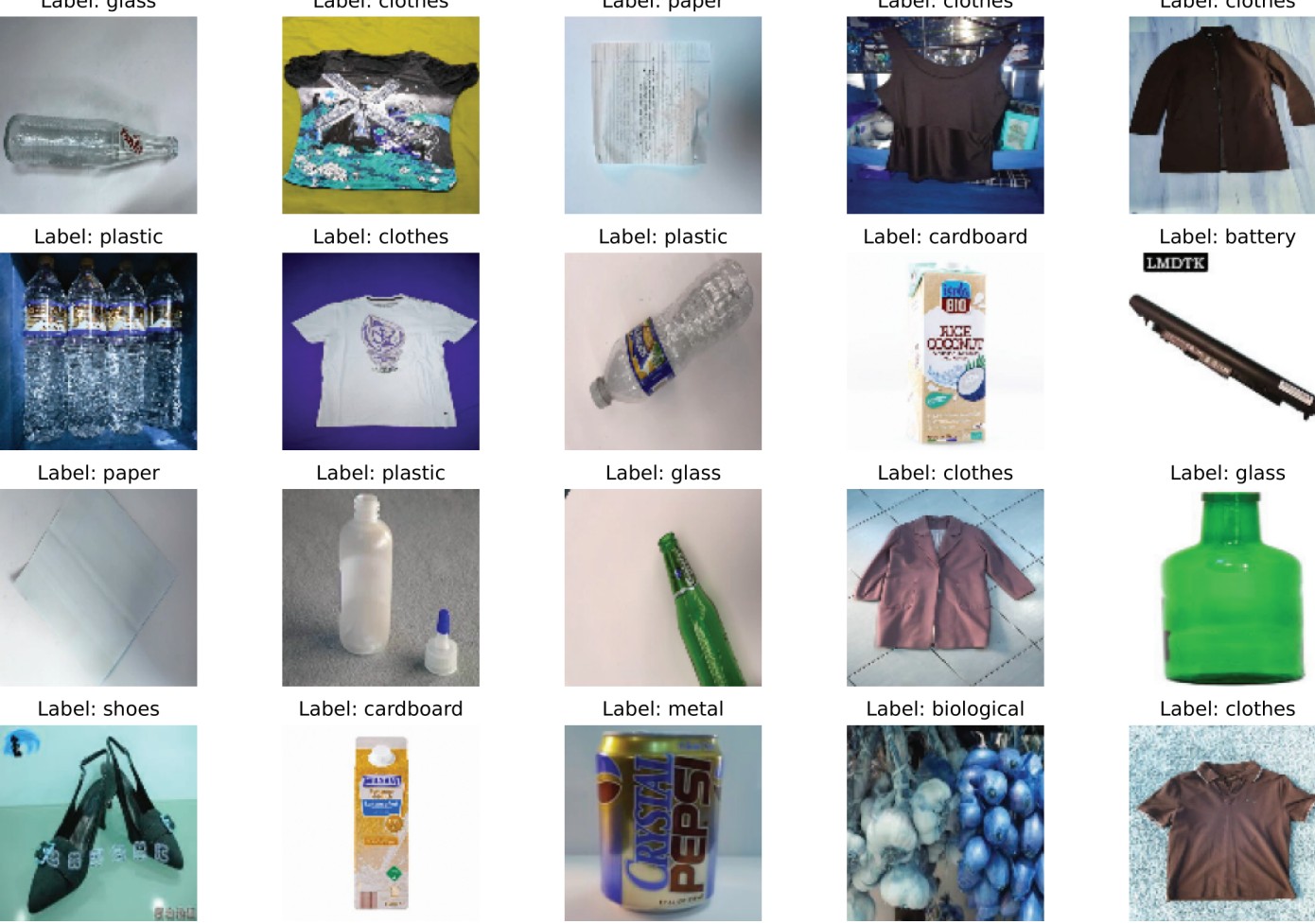

**Fig 1. Sample images from the four distinct waste material datasets.**

**Table 1. An overview of the datasets.**

| Datasets | No. of Classes | No. of Images |
|---|---|---|
| Waste classification V2 [28] | 10 | 21983 |
| Waste classification [29] | 12 | 15515 |
| OpenRecycle [30] | 7 | 2301 |
| TrashNet [31] | 6 | 2527 |
| Waste Images [33] | 9 | 8346 |
| Recyclable Waste Images [34] | 9 | 3180 |
| RealWaste [36] | 9 | 4753 |

**Table 2. Unique waste classes throughout the 7 datasets.**

| Sl. No. | Item | Sl. No. | Item | Sl. No. | Item |
|---|---|---|---|---|---|
| 1 | Aluminium | 10 | E-waste | 19 | Organic waste |
| 2 | Battery | 11 | Egg packaging | 20 | Crumpled paper |
| 3 | Biological | 12 | Foil | 21 | Paper |
| 4 | Brown-glass | 13 | Food organics | 22 | PET plastics |
| 5 | Cardboard | 14 | Glass Bottle | 23 | Receipt |
| 6 | Carton | 15 | Wood | 24 | Shoes |
| 7 | Clothes | 16 | Green-glass | 25 | Textiles |
| 8 | Other plastics | 17 | Metal | 26 | Vegetation |
| 9 | Disposable paper cups | 18 | Miscellaneous trash | 27 | White-glass |

carefully selected to provide a diverse yet representative cross-section of the most commonly encountered waste types globally. Notably, the World Bank's "What a Waste 2.0" [2] report identifies food and green waste, plastics, paper, cardboard, metals, glass, wood, and textiles as the core components of municipal solid waste (MSW)—all of which are present in the selected datasets—collectively accounting for over 80% of waste in many urban regions. Furthermore, the 2015 UNEP Global Waste Management Outlook [4] supports the classification strategy by highlighting the growing environmental and health risks associated with categories such as e-waste, food organics, and plastic waste, all of which are explicitly included in the datasets. This variety is essential for assessing the effectiveness of the proposed model, as it facilitates a thorough evaluation of its capacity to yield favorable outcomes across different data distributions and waste categories.

# 4 Research methodology

The research methodology initiates with resizing images to $170 \times 170$ pixels for datasets with a greater number of images or $290 \times 290$ pixels for smaller datasets. This deliberate choice of larger image sizes for smaller datasets aims to preserve intricate details, mitigate potential information loss, and foster model generalization, thus contributing to improved results without necessitating additional data augmentation techniques. To better simulate real-world conditions where image quality may be inconsistent, we trained the model on raw, unprocessed images, ensuring consistency across datasets and aligning with neural network architectures for optimal performance. This strategy helps to mitigate potential variability across datasets while allowing the model to effectively learn from diverse, real-world data.

A sophisticated waste classification model is then crafted, blending TL-based DenseNet201 with a customized neural network architecture. Leveraging the capabilities of a pre-trained DenseNet201 model facilitates the extraction of intricate image features. These features undergo refinement through dual branches, with one branch utilizing MaxPooling and the other AveragePooling, while incorporating Squeeze-and-Excitation (SE) blocks for enhanced feature representation. Then global average pooling is employed to reduce spatial dimensions, and final classification is conducted through fully connected dense layers, yielding class probabilities. This fusion of transfer learning and custom architecture empowers precise image classification, leveraging a rich array of learned features. Additionally, Grad-CAM can be integrated to provide visual explanations of the model's decision-making process, enhancing interpretability and trust in the results.

The research culminates in a website capable of detecting waste in uploaded images. This platform bridges the gap between theoretical research and practical application by leveraging advanced algorithms to identify and categorize waste effectively. By enabling consumers to handle garbage more effectively, it seeks to solve environmental problems and contribute to a sustainable future. Finally, potential areas for future research have been pinpointed. These upcoming avenues are geared towards enhancing waste management technologies and playing a greater role in promoting a sustainable future.

## 4.1 Customized DenseNet201 in fusion with squeeze and excitation (SE) integrated parallel CNN architecture

In the quest to enhance image classification capabilities, this methodology relies on the well-established DenseNet201 architecture, renowned for its proficiency in analyzing complex image attributes, initially trained on the ImageNet dataset. Rather than initiating training from scratch, the approach opts for fine-tuning, a process that capitalizes on the already learned features stored in the pre-existing weights derived from ImageNet. This methodological choice not only streamlines the training process but also ensures efficiency by leveraging the rich knowledge embedded in the pre-trained weights, thereby streamlining the training process and enhancing efficiency.

**Fine tuning of DenseNet201 architecture:** The significance of fine-tuning cannot be overstated, especially considering the substantial depth of the DenseNet201 architecture. The discerning strategy of selective fine-tuning involves retaining the pre-trained weights while freezing updates for all convolutional layers except the final ones. This strategic decision helps mitigate the risk of overfitting by directing the focus of training efforts towards the latter convolutional layers. Meanwhile, adjustments are meticulously applied to the upper layers of the architecture, intensifying the training emphasis on these layers to capture nuanced and high-level features relevant to the specific image classification task. For a visual representation, Fig 2 illustrates the entire architecture, highlighting the fine-tuned DenseNet201 model tailored for waste classification.

**Adaptive convolutional layers on top of DenseNet201:** During the fine-tuning process, the convolutional layers within DenseNet201 undergo a meticulous adaptation to maintain their excellence in extracting spatial information from input images. The diverse kernel sizes, such as 7x7 and 5x5, remain integral, finely tuning their ability to capture hierarchical patterns at various scales. This adaptability proves crucial, particularly in the nuanced task of discerning both broad contextual information and finer details. The integration of Rectified Linear Unit (ReLU) activation functions persists, introducing non-linearity and enabling the model to refine its understanding of complex patterns within the refined dataset adeptly.

Additionally, Batch Normalization continues to play a pivotal role, steadfastly stabilizing and expediting the fine-tuning process. By normalizing layer inputs, it mitigates internal covariate shifts and provides essential regularization. This thoughtful and adaptive combination ensures the model's continued proficiency in navigating the intricate landscape of feature transformation, further solidifying its capability in comprehending and categorizing images, even as it undergoes the iterative process of fine-tuning.

**Parallel CNN branches for feature map processing:** The narrative unfolds into two parallel branches, each equipped with a distinct approach to processing feature maps. In the first branch, max-pooling layers are employed with a stride of (2, 2), effectively downsampling feature maps while preserving essential information. Conversely, the second branch utilizes average-pooling layers, contributing to a diverse set of feature reduction techniques. In both

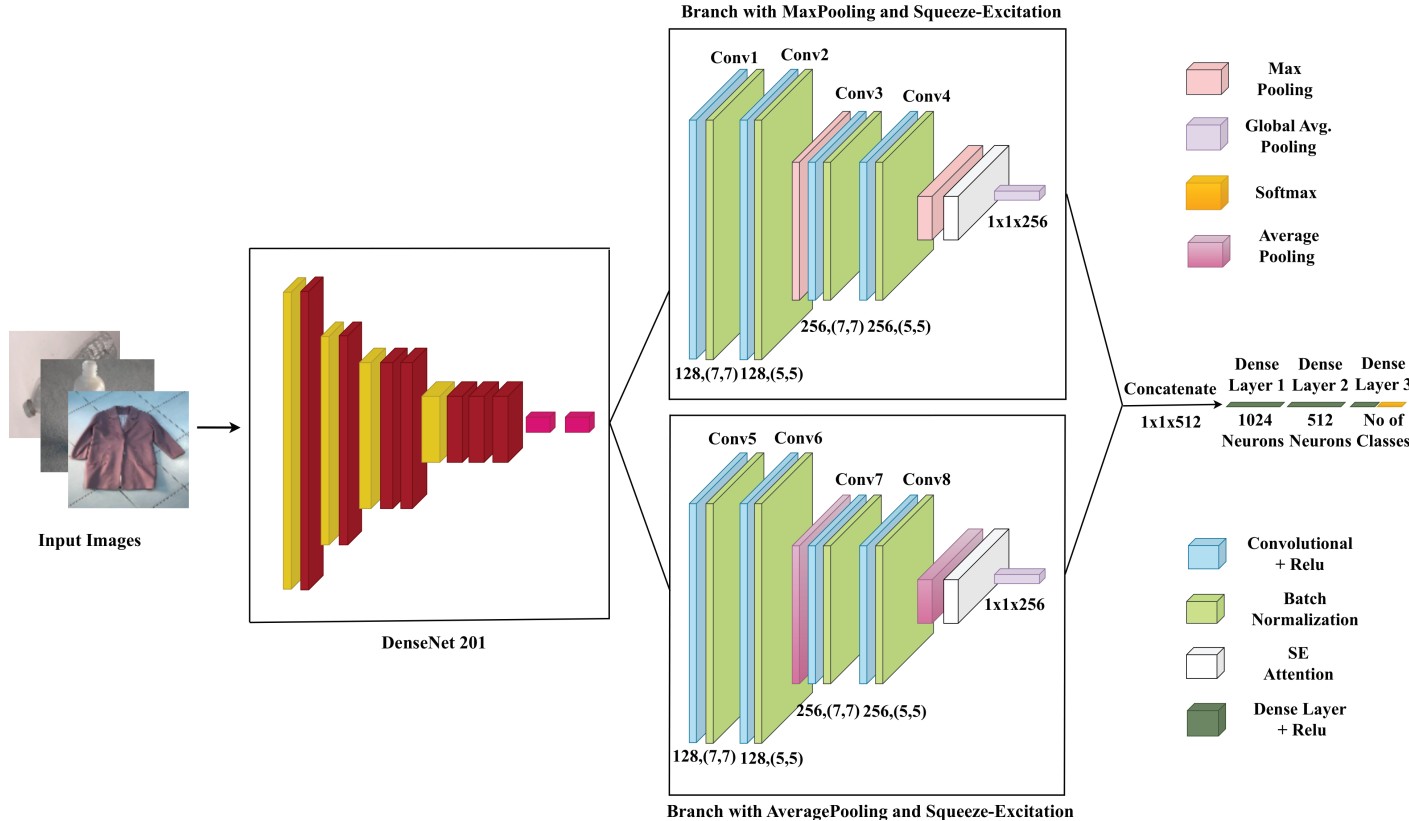

**Fig 2. Customized DenseNet-201 with Squeeze and Excitation Integrated Parallel CNN Architecture.**

branches, convolutional layers maintain filter sizes of 7x7 and 5x5, ensuring the integrity of spatial information within the feature maps. The incorporation of SE attention blocks in each branch enhances the richness of information within the feature maps. Following this, global average pooling reduces spatial dimensions to (1, 1), preparing for the consolidation of feature maps from both branches.

**Squeeze and Excitation (SE) attention block:** To enhance the model's feature representation, the Squeeze-and-Excitation (SE) attention block is skillfully integrated. This block, meticulously designed to amplify the importance of feature maps, operates through global average pooling. Subsequently, the feature maps undergo calibration, orchestrated by learnable weights within densely connected layers, activated by ReLU. These architectural refinements endow the model with the ability to emphasize the most significant regions within the feature maps, thereby enhancing its discernment capability. Finally, the feature maps generated from two parallel branches are concatenated to produce the most informative and diverse features.

**Final dense layers and classification:** In the final phase, the model uses a series of dense layers. These layers start with 1024 units, then go down to 512 units, and end with a layer that has neurons matching the total number of categories for classification. During this phase, the model looks for complex patterns and relationships in the features it has learned. The outputs from the two branches, each with their unique insights, are combined. This combination helps in the next step where the model puts together what it has learned from both branches.

**Justification of using parallel CNN on top of DenseNet201 architecture:** In the realm of CNN, the adoption of a parallel approach, notably when coupling one branch with max pooling and another with average pooling, unveils a refined strategy for image processing. This duality of pooling methods imparts a distinctive advantage, orchestrating a harmonious synthesis of salient and subtle features inherent in the input data. Max pooling, with its prowess in accentuating high-impact patterns, stands alongside average pooling, which gracefully gauges the broader distribution of information.

The confluence of these complementary methodologies imparts a profound sophistication to the model, fostering a more profound and nuanced comprehension of intricate details and hierarchical relationships within the dataset. This parallel architecture, seamlessly attuned to the simultaneous extraction of diverse information facets, transcends conventional models. Its capacity for rendering a richer representation of input data often results in heightened levels of generalization and superior classification efficacy, surpassing models predicated upon singular pooling methodologies or devoid of parallel intricacies.

## 4.2 Squeeze and Excitation (SE) attention mechanism

The Squeeze and Excitation (SE) attention mechanism operates by reducing the influence of less relevant features while enhancing the importance of crucial ones. This process occurs before the final fully connected layers, where the model learns and updates weights through back-propagation. Fig 3 presents a visual representation of how the SE attention mechanism works sequentially. It effectively demonstrates how the model dynamically shifts its attention to highlight significant features during the classification process. A brief description of the SE attention mechanism is given below:

**Squeeze phase (global average pooling):** The goal of the squeeze phase is to condense spatial information and create a channel-wise summary. Global Average Pooling (GAP) is applied across the spatial dimensions, computing the average activation value for each channel.

$$z = \frac{1}{H \times W} \sum_{i=1}^{H} \sum_{j=1}^{W} x_{i,j,k} \tag{1}$$

Where, $H$ and $W$ are the height and width of the input tensor, and $x_{i,j,k}$ represents the activation at position $(i, j)$ in channel $k$.

**Excitation phase (fully connected layers):** The excitation phase focuses on capturing channel-wise dependencies. It employs two fully connected layers to model the relationships between channels. The first layer reduces the number of channels by a factor of $\frac{1}{se\_ratio}$ with a ReLU activation, and the second layer restores the original number of channels with a sigmoid activation.

$$y = \text{ReLU}\left(\text{FC}\left(z, \frac{C}{se\_ratio}\right)\right) \tag{2}$$

$$s = \text{Sigmoid}(\text{FC}(y, C)) \tag{3}$$

Where, $C$ is the number of channels, FC denotes the fully connected layer, and $se\_ratio$ is the channel-wise squeeze factor.

**Scale and excite (element-wise multiplication):** This step combines the original input tensor with the channel-wise squeeze factor obtained from the excitation phase. This

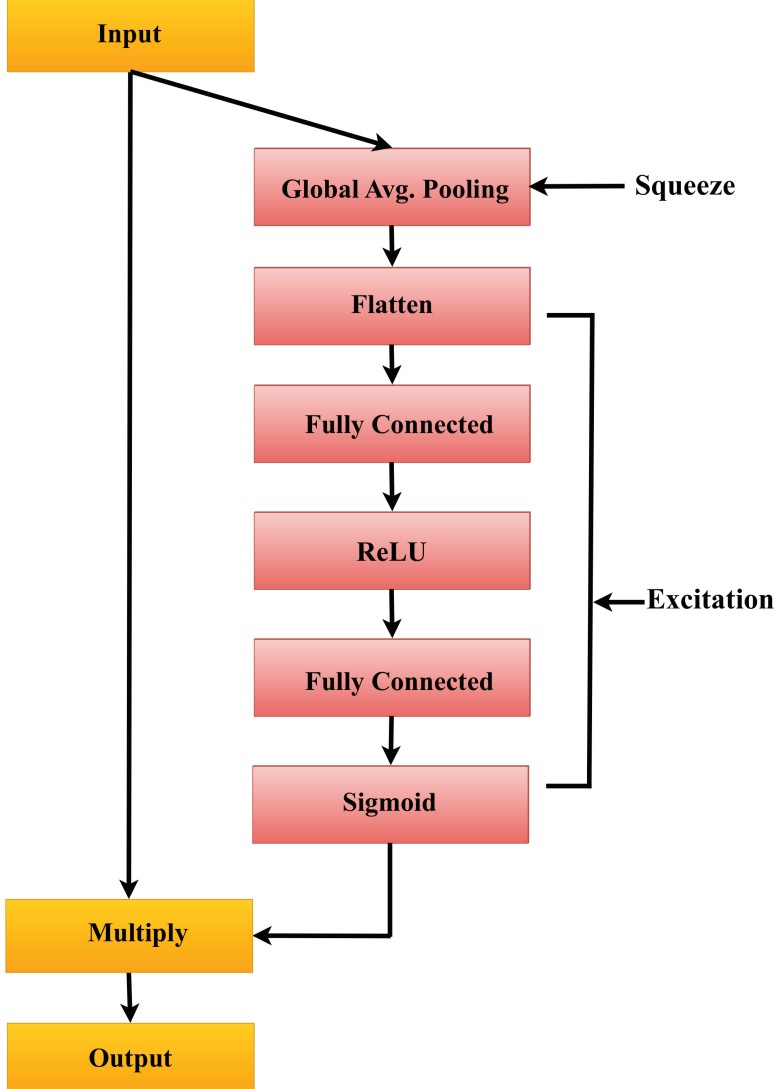

**Fig 3. Squeeze and Excitation Mechanism.**

element-wise multiplication scales the input tensor based on the importance assigned to each channel.

$$output = x \odot s, \tag{4}$$

where $\odot$ denotes element-wise multiplication.

**Justification of using Squeeze and Excitation (SE) mechanism in customized DenseNet201 architecture:** Within the modified DenseNet201 architecture, the inclusion of the SE attention block enables the network to dynamically prioritize or de-prioritize specific channels. This adaptive adjustment empowers more efficient feature learning and representation by intensifying (exciting) the most crucial features while compressing (squeezing) those considered less essential. Consequently, this process boosts the discriminative capability of the features, enabling the model to concentrate on the most relevant aspects of the data. As a

result, classification performance is enhanced as attention is directed towards the most salient features.

## 4.3 Performance evaluation measures

To evaluate the performance of our proposed model, several well-known statistical evaluation measures such as Accuracy, Precision, Recall, F1 Score, Specificity,Receiver Operating Characteristic (ROC) Curve, Matthews Correlation Coefficient (MCC), Cohen's Kappa, and Geometric Detection Rate (GDR) were considered. The details of each evaluation measure are given below:

**Accuracy:** Accuracy measures the proportion of correctly classified instances out of the total instances.

$$\text{Accuracy} = \frac{TP + TN}{TP + TN + FP + FN} \tag{5}$$

**Precision:** Precision measures the proportion of true positive predictions out of all positive predictions.

$$\text{Precision} = \frac{TP}{TP + FP} \tag{6}$$

**Recall (sensitivity or true positive rate):** Recall measures the proportion of true positive predictions out of all actual positive instances.

$$\text{Recall} = \frac{TP}{TP + FN} \tag{7}$$

**F1 Score:** F1 Score is the harmonic mean of precision and recall, providing a balance between the two.

$$\text{F1 Score} = \frac{2 \cdot \text{Precision} \cdot \text{Recall}}{\text{Precision} + \text{Recall}} \tag{8}$$

**Specificity (true negative rate):** Specificity measures the ability of a classification model to correctly identify negative instances.

$$\text{Specificity} = \frac{TN}{TN + FP} \tag{9}$$

A high specificity indicates that the model is good at avoiding false alarms or false positives.

**Matthews Correlation Coefficient (MCC):** MCC is a balanced measure of binary classification performance, considering all four confusion matrix categories (TP, TN, FP, FN). It ranges from -1 (indicating inverse prediction) to +1 (indicating perfect prediction), with 0 representing random prediction.

$$\text{MCC} = \frac{(TP \cdot TN) - (FP \cdot FN)}{\sqrt{(TP + FP) \cdot (TP + FN) \cdot (TN + FP) \cdot (TN + FN)}} \tag{10}$$

**Cohen's Kappa:** Cohen's Kappa measures the agreement between two raters or classifiers, accounting for the possibility of agreement occurring by chance. It ranges from -1 (complete disagreement) to +1 (perfect agreement), with higher values indicating stronger agreement.

$$\text{Kappa} = \frac{P_o - P_e}{1 - P_e} \tag{11}$$

where:

- Po (Observed Agreement) is the proportion of times the raters agree. It is computed as:

$$P_o = \frac{TP + TN}{TP + TN + FP + FN} \tag{12}$$

- Pe (Expected Agreement) is the proportion of agreement expected by chance. It is computed based on the marginal totals of the confusion matrix as:

$$P_e = \left( \frac{(TP + FP) \cdot (TP + FN) + (TN + FP) \cdot (TN + FN)}{(TP + TN + FP + FN)^2} \right) \tag{13}$$

**Geometric Detection Rate (GDR):** GDR evaluates the model's overall ability to correctly identify true positives and true negatives, offering a balanced view of performance across all classes. It combines both sensitivity (recall) and specificity.

$$GDR = \sqrt{\frac{\sum_i TP_i}{\sum_i (TP_i + FN_i)} \cdot \frac{\sum_i TN_i}{\sum_i (TN_i + FP_i)}} \tag{14}$$

where:

- $TP$ (True Positives): The number of correctly predicted positive instances.
- $TN$ (True Negatives): The number of correctly predicted negative instances.
- $FP$ (False Positives): The number of actual negatives incorrectly classified as positives.
- $FN$ (False Negatives): The number of actual positives incorrectly classified as negatives.

**Receiver Operating Characteristic (ROC) curve:** ROC curve is a graphical representation of a binary classification model's performance. It plots the True Positive Rate (TPR) against the False Positive Rate (FPR) at various threshold settings. ROC AUC (Area Under the Curve) is a single-value measure of the ROC curve's performance.

## 4.4 Experimental setup

The experiment was conducted on Kaggle using the TensorFlow framework. A single NVIDIA Tesla T4 GPU with 16GB RAM was utilized for parallel processing during training. The Adam optimizer, configured with a learning rate of 0.001, was employed to balance convergence efficiency and model stability. To address overfitting, early stopping was implemented, monitoring validation loss and stopped training when the validation loss did not improve for 25 epochs with a minimal delta of 0.001. Table 3 offers a concise summary of crucial parameters and configurations, aiding in comprehending the experimental setup.

## 5 Result analysis

The analysis and discussion of experimental results are divided into several phases. It begins with findings from experiments on four primary waste material classification datasets, followed by an evaluation of robustness under adversarial attacks and environmental variability. To validate these results, validation across three additional datasets is incorporated. Next, the performance of the customized DenseNet201, enhanced with Squeeze and Excitation (SE) blocks, is compared against various established CNN architectures using identical training, validation, and test sets, as well as traditional machine learning classifiers, to ensure fair

**Table 3. Experimental setup for the proposed classification model.**

| Parameter | Value |
| --- | --- |
| Framework | TensorFlow (Python-3) |
| GPU | NVIDIA Tesla T4 |
| GPU Memory | 16GB RAM |
| Data Split | 70:10:20 |
| Activation Function | ReLU |
| Output Activation Function | Softmax |
| Optimizer | Adam |
| Learning Rate | 0.001 |
| Batch Size | 10 |
| Overfitting Mitigation | Early stopping (25 epochs, delta = 0.001) |

assessment. The performance across seven datasets is rigorously evaluated against state-of-the-art (SOTA) studies, offering a comprehensive comparison in terms of efficacy and innovation. Additionally, the study presents an overview of empirical time complexity, energy consumption, and a detailed benchmarking of model complexity and deployment viability across all datasets. Grad-CAM visualizations are also incorporated to enhance interpretability.

## 5.1 Performance analysis of waste classification V2 dataset

The accuracy and loss curves depicted in Fig 4 provide a detailed insight into the performance of the DenseNet201-infused SE integrated Parallel CNN model on the Waste Classification V2 dataset. This graphical representation offers extensive insights into the capabilities of the model and is very helpful in understanding its learning processes and performance features.

Compared to the validation set's 91.00% accuracy and lower loss, the model's early training results indicated an 80.00% training accuracy and a greater training loss. Backpropagation updates decreased training loss and increased training accuracy as training progressed. There was a notable decrease in validation loss and an approximate 97.00% validation accuracy. Overall, the model's learning and generalization were effectively demonstrated by the steadily

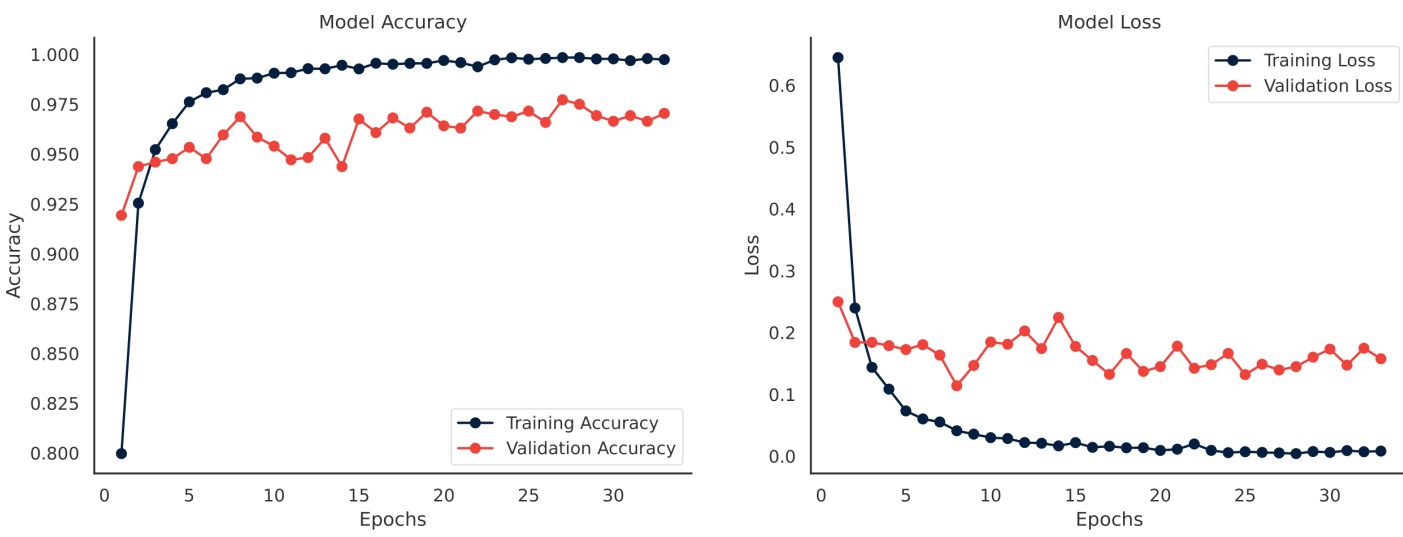

**Fig 4. Accuracy & loss curves of the DenseNet201-Infused Parallel SE-CNN for Waste Classification V2 dataset.**

rising accuracy curves, while ongoing error minimization and refinement were revealed by the continuously declining loss curves.

Training accuracy and loss showed a slight improvement after the tenth epoch, however, validation measures showed slight variations as a result of weight updates brought about by backpropagation. An early termination parameter was introduced to evaluate validation loss over a 25 epochs period due to concerns over possible overfitting. The model reached its maximum training accuracy of 99.81% by stopping training at the 33rd epoch, while its highest validation accuracy was 97.00% following the ninth epoch. The model's optimal performance was captured by saving the weights from the ninth epoch for future testing on an independent test set, and the early halting strategy prevented overfitting. The consistent improvement in model accuracy and decrease in loss metrics indicate that the model is learning and optimizing its performance effectively over the epochs.

A confusion matrix is a tabular representation of a classification model's performance, displaying the number of true positives, true negatives, false positives, and false negatives. It provides insights into the model's accuracy and errors by comparing predicted and actual classifications. Fig 5 depicts the performance of the proposed model for the independent test set. The rows of the matrix represent the true labels of the images, and the columns represent the

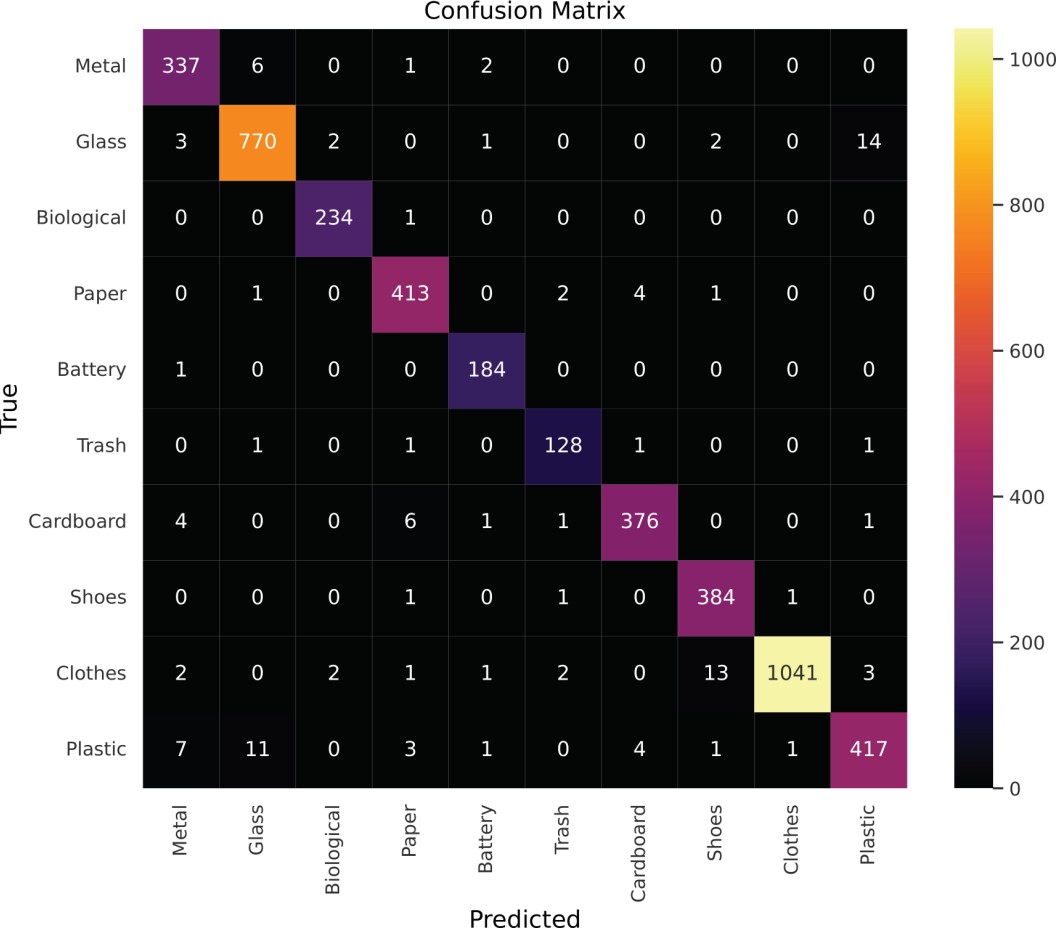

**Fig 5. Confusion matrix for Waste Classification V2 dataset.**

predicted labels. The diagonal entries of the matrix represent the number of images that were correctly classified, while the off-diagonal entries represent the number of images that were misclassified.

The experimental results analysis revealed that the model generally performed well in classifying Biological, Battery, Trash, and Shoes materials as evidenced by highly correct classifications and minimal misclassifications into other categories. However, it encountered difficulties in distinguishing between certain classes, such as Metal and Glass, Paper and Cardboard, as well as Paper and Plastic. This was evident from the higher numbers of misclassifications observed in these instances. Specifically, there was notable confusion between materials like Metal and Glass where 14 Glass samples were predicted as Plastic and 11 samples were predicted as Glass. This highlights that the proposed model struggled a bit to distinguish between Metal and Glass. On the other hand, the proposed model also wrongly predicted 13 Clothes samples as Shoes.

Table 4 presents a comprehensive analysis of performance across various classes calculated from the confusion matrix using the Waste Classification V2 dataset. Precision scores, which gauged the accuracy of positive predictions, were consistently high, ranging from 95% to 100%, indicating minimal false positives. Moreover, recall scores, measuring the model's ability to capture true positives, also exhibited strong performance, with values ranging from 94% to 100%, suggesting effective identification of most instances of each waste class. The F1 scores, representing a balance between precision and recall, were notably high, ranging from 95% to 99%, affirming the model's overall robustness in classification tasks. Additionally, specificity scores reflected the model's proficiency in identifying true negatives, ranging from 94% to 100%, indicating minimal misclassification of other materials as the specified waste types. According to the table, no noticeable bias is observed in the model's performance across the various classes.

Fig 6 presents the uniformity of precision, recall, F1 score, specificity, accuracy, MCC, Cohen's Kappa, and Geometric Detection Rate (GDR) at 0.97 across all metrics, suggesting a highly consistent and reliable performance of the waste classification model. This level of performance indicates that the model effectively minimizes both false positives and false negatives, achieving a balance between precision and recall. The high F1 score further confirms the model's ability to maintain a harmonious trade-off between precision and recall, ensuring robustness in classification tasks. Additionally, the specificity score at 97.00% highlights the model's proficiency in correctly identifying true negatives, thereby reducing the risk of misclassifying other materials as specified waste types. The stable MCC and Kappa scores of 0.97 demonstrate a robust alignment between the predicted and actual classifications, thereby

**Table 4. Classwise metrics overview for Waste classification V2 dataset.**

| Class Name | Metrics (%) | | | |
|---|---|---|---|---|
| | Precision | Recall | F1 Score | Specificity |
| Metal | 95.00 | 97.00 | 96.00 | 97.00 |
| Glass | 98.00 | 97.00 | 97.00 | 97.00 |
| Biological | 98.00 | 100 | 99.00 | 100 |
| Paper | 97.00 | 98.00 | 97.00 | 98.00 |
| Battery | 97.00 | 99.00 | 98.00 | 99.00 |
| Trash | 96.00 | 97.00 | 96.00 | 97.00 |
| Cardboard | 98.00 | 97.00 | 97.00 | 97.00 |
| Shoes | 96.00 | 99.00 | 97.00 | 99.00 |
| Clothes | 100 | 98.00 | 99.00 | 98.00 |
| Plastic | 96.00 | 94.00 | 95.00 | 94.00 |

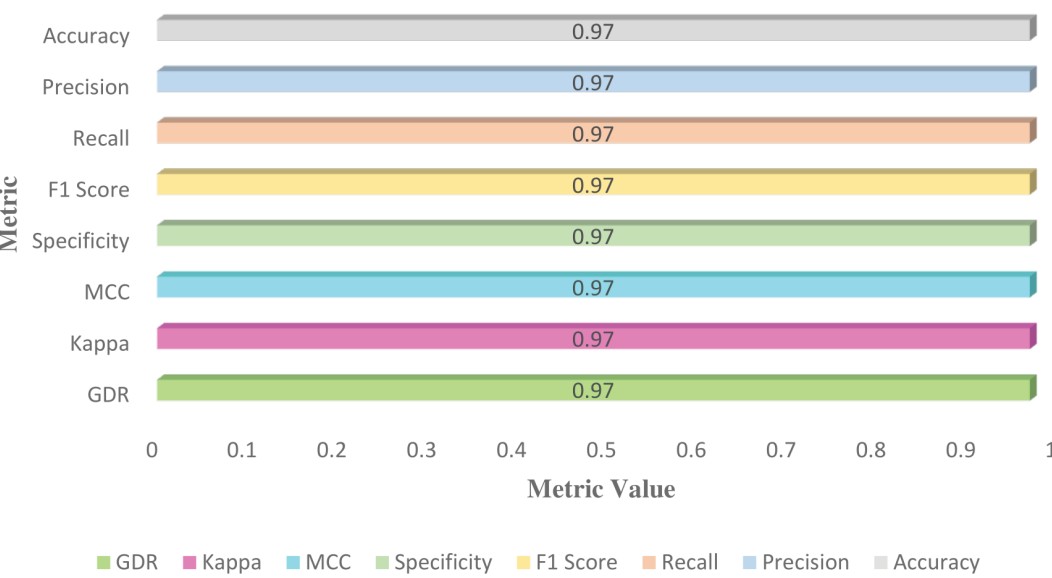

Fig 6. Overall Metrics overview for Waste Classification V2 dataset.

confirming the model's precision. Additionally, the GDR score, which also stands at 97.00%, highlights the model's comprehensive ability to accurately recognize both true positives and true negatives.

Fig 7 shows a receiver operating characteristic (ROC) curve for waste classification for Waste Classification V2 dataset. An ROC curve is a graph that shows the performance of a binary classifier at different thresholds. The x-axis of the ROC curve shows the false positive rate (FPR), which is the proportion of negative examples that are incorrectly classified as positive. The y-axis of the ROC curve shows the true positive rate (TPR), which is the proportion of positive examples that are correctly classified as positive. The ideal ROC curve is a straight line that goes from the bottom left corner (TPR=0, FPR=0) to the top left corner (TPR=1, FPR=0). In practice, ROC curves are usually not perfectly linear, but the closer they are to the ideal curve, the better the model's performance. The ROC curve demonstrated excellent performance across all classes, with an Area Under the Curve (AUC) of 1.00 for each class. This indicated that the model had perfect discrimination ability for distinguishing between different types of waste. The high AUC values suggested that the model had a high true positive rate and a low false positive rate, making it highly reliable in classifying waste items.

## 5.2 Performance analysis of waste classification dataset

The accuracy and loss curves in Fig 8 provide a thorough overview of the performance of the DenseNet201 infused SE integrated Parallel CNN model on the waste classification dataset. In contrast to the validation set, which attained an accuracy of 88.50% and a lower loss for the first epoch, the model first showed a training accuracy of 78.00% with a larger training loss. Training accuracy and training loss decreased as a result of backpropagation as training went on. A notable reduction in validation loss was seen, together with an increase in validation

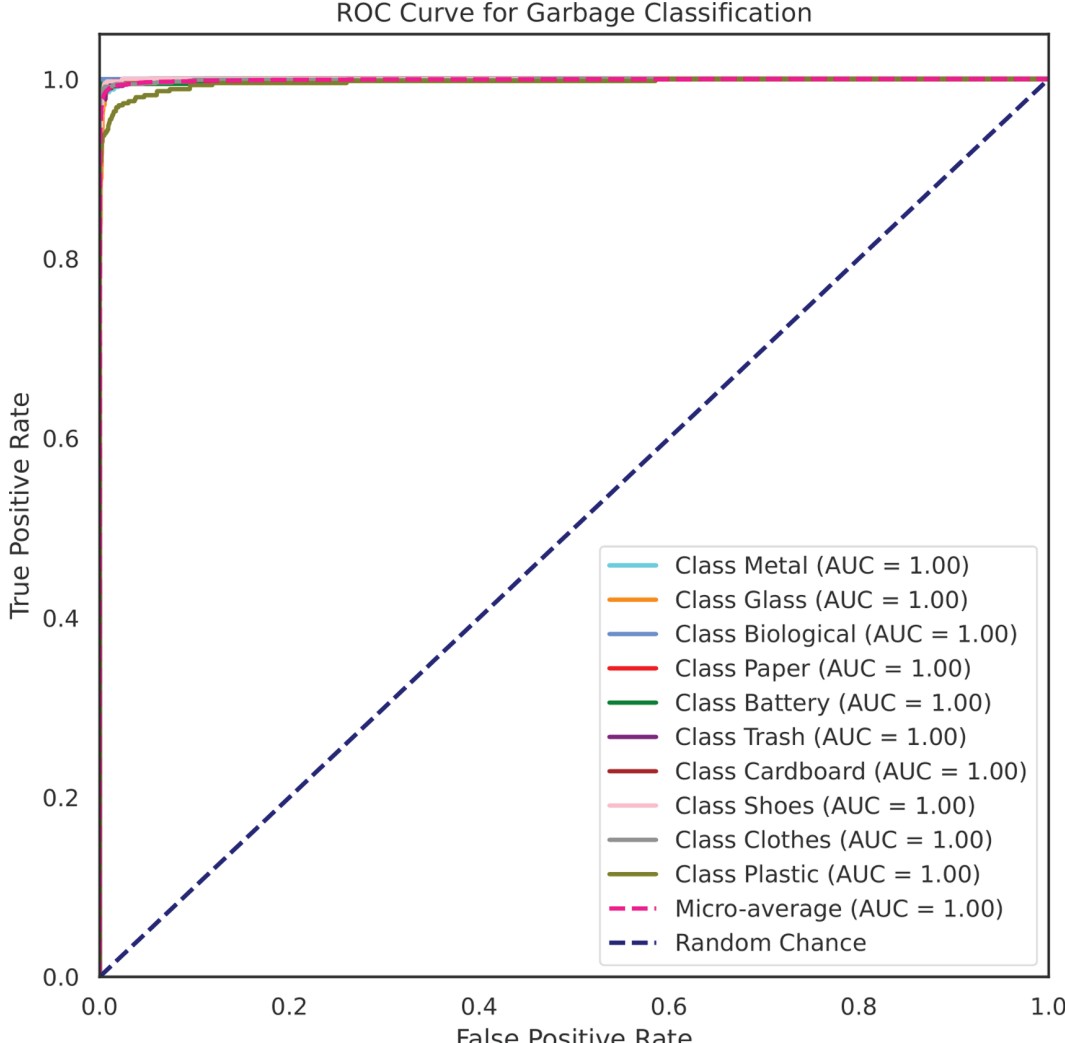

**Fig 7. ROC curve for Waste Classification V2 dataset.**

accuracy to around 96.50%. For both sets, the accuracy curve steadily increased, demonstrating the model's strong learning and generalization ability. Concurrently, the training set's loss curve showed a gradual decrease, while the validation set showed slight variations.

Training accuracy and loss plateaued at the tenth epoch, but validation measures exhibited slight variations as a result of backpropagation updates. There were worries about possible overfitting due to the prolonged absence of discernible training improvement. The accuracy curve showed saturation at about the 40th epoch, indicating diminishing returns from the training set. As the model got closer to its ideal condition, the loss curve likewise reached a plateau. After 40 epochs, more training might not produce much of an improvement and might even cause overfitting. The accuracy graph showed an astounding growth to 99.92% over 40 epochs, highlighting the model's proficiency in categorizing training instances with 99.92% accuracy and respectable validation accuracy of 97.26% peaked at the $29^{th}$ epoch. The model exhibits successful learning as it continuously increases accuracy and decreases loss.

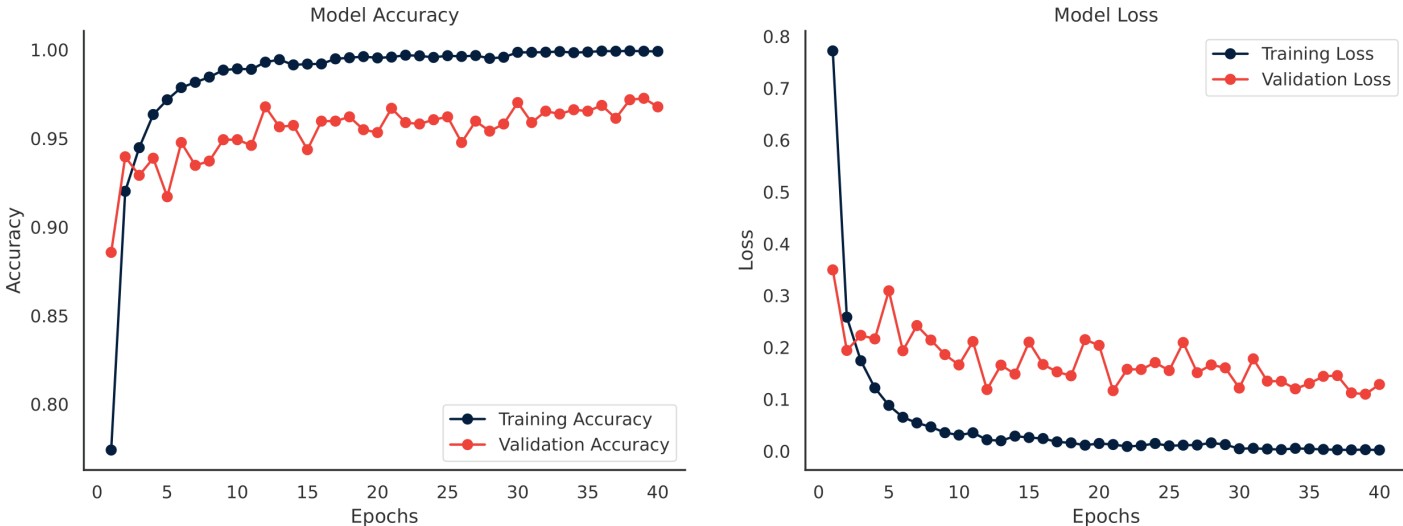

**Fig 8. Accuracy & loss curves of the DenseNet201-Infused Parallel SE-CNN for Waste Classification dataset.**

The confusion matrix for the proposed model, shown in Fig 9, illustrates its performance on the independent test set, detailing the classification results across 12 classes in the waste classification dataset. Sparse non-diagonal elements highlight overall effectiveness, with notable misclassifications such as Clothes as Shoes (12 occurrences) and Plastic as Metal (4 occurrences), offering insights for model refinement in this multi-class setting. The analysis of experimental results indicates the model's strong performance in correctly classifying Biological, Paper, Clothes, and Green glass, with minimal misclassifications into other categories.

A thorough performance study utilizing the Waste Classification dataset is shown in Table 5 for 12 classes. Precision scores, which evaluate the degree of accuracy in positive predictions, have regularly fallen between 90.00% and 100.00%, suggesting a low rate of false positives. Recall scores, which gauge how well the model can identify true positives, performed well (89.00% to 99.00%), indicating that the majority of instances for each waste class were effectively identified. The model's general resilience in classification tasks was confirmed by the exceptionally high F1 scores (91.00% to 99.00%), which indicate a balance between precision and recall. Specificity scores, which ranged from 89.00% to 99.00% and indicated a limited amount of misclassification of other materials as the designated waste kinds, highlighted the model's ability to recognize genuine negatives.

Precision, recall, F1-score, specificity, and accuracy all uniformly register at 97.00% in Fig 10, demonstrating the waste classification model's consistently dependable performance. The model's capacity to maintain a harmonious trade-off between precision and recall is highlighted by the high F1 score. The model's ability to accurately detect real negatives is demonstrated by its specificity score of 97.00%, which lowers the possibility of misclassifying other materials. Furthermore, the MCC, Kappa, and GDR scores, each at 0.96, further validate the model's strength. The MCC and Kappa scores demonstrate a high level of concordance between the predicted and actual classifications, while the GDR illustrates the model's overall proficiency in accurately identifying both true positives and true negatives.

The ROC curve, illustrated in Fig 11, visually represents a binary classifier's performance by plotting the false positive rate (FPR) against the true positive rate (TPR). For the

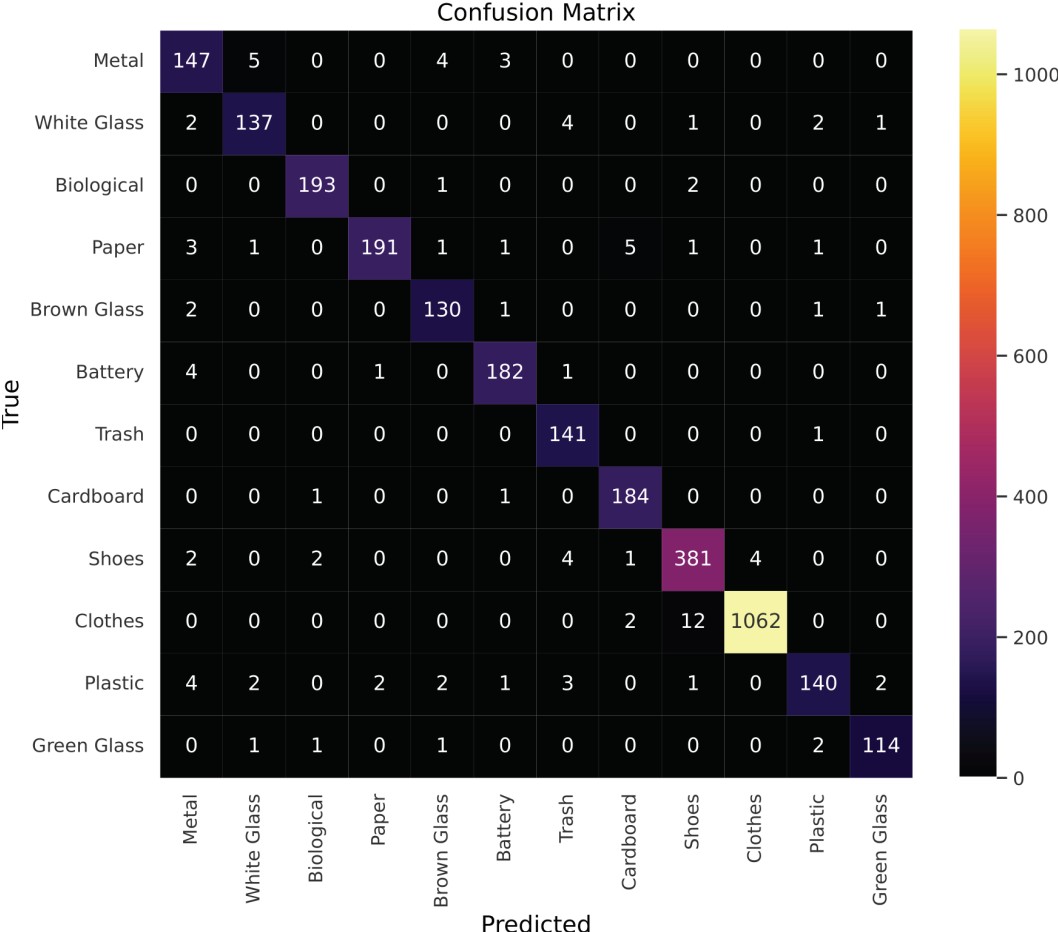

**Fig 9. Confusion Matrix for Waste Classification dataset.**

**Table 5. Classwise metrics overview for the waste classification dataset.**

| Class Name | Metrics (%) | | | |
|---|---|---|---|---|
| | Precision | Recall | F1 Score | Specificity |
| Metal | 90.00 | 92.00 | 91.00 | 92.00 |
| White Glass | 94.00 | 93.00 | 94.00 | 93.00 |
| Biological | 98.00 | 98.00 | 98.00 | 98.00 |
| Paper | 98.00 | 94.00 | 96.00 | 94.00 |
| Brown Glass | 94.00 | 96.00 | 95.00 | 96.00 |
| Battery | 96.00 | 97.00 | 97.00 | 97.00 |
| Trash | 92.00 | 99.00 | 96.00 | 99.00 |
| Cardboard | 96.00 | 99.00 | 97.00 | 99.00 |
| Shoes | 96.00 | 97.00 | 96.00 | 97.00 |
| Clothes | 100 | 99.00 | 99.00 | 99.00 |
| Plastic | 95.00 | 89.00 | 92.00 | 89.00 |
| Green Glass | 97.00 | 96.00 | 96.00 | 96.00 |

Waste Classification dataset, the ROC curve in Fig 11 exhibits an exemplary area under the curve (AUC) of 1.0 for all classes, except for plastic. This exceptional AUC signifies the model's remarkable ability to accurately differentiate between positive and negative examples across the majority of categories. In the case of the plastic class, the classifier achieves a

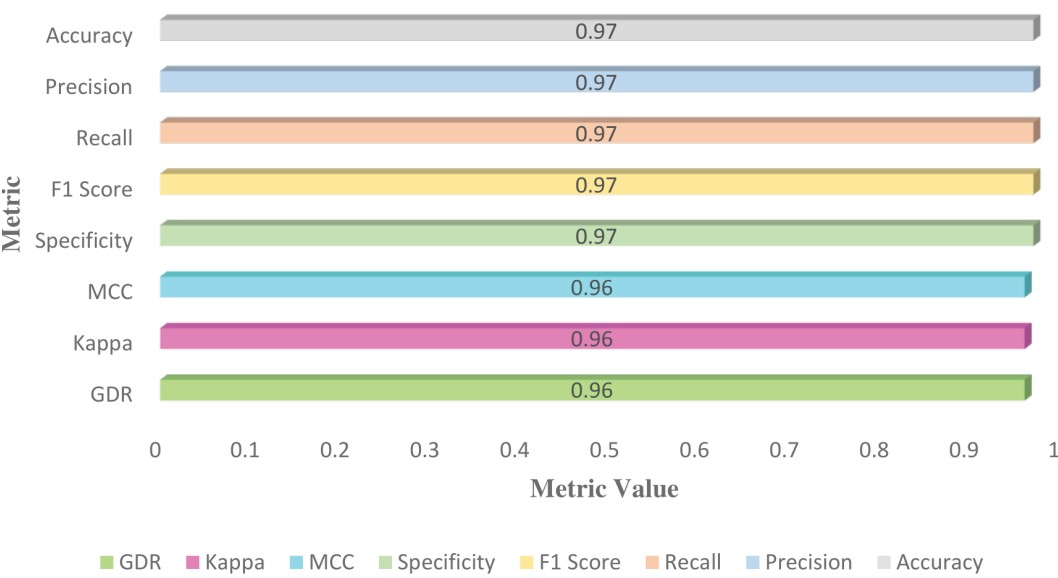

**Fig 10. Overall Metrics overview for Waste Classification dataset.**

commendable AUC of 0.99, indicating high proficiency in distinguishing between positive and negative instances, albeit with a minor occurrence of errors.

### 5.3 Performance analysis of OpenRecycle dataset

The accuracy and loss curves in Fig 12 provide a thorough understanding of the performance of the DenseNet201-infused SE integrated Parallel CNN model on the OpenRecycle classification dataset. The model initially showed a training accuracy of 59.00% with a larger training loss, in contrast to the validation set, which reached 81.00% accuracy and reduced loss. Backpropagation caused training accuracy and loss to decrease as training went on. Around the 34th epoch, there was a notable decrease in validation loss and an increase in validation accuracy to roughly 95.00%. Both accuracy curves rose steadily, demonstrating the model's good capacity for learning and generalization. Concurrently, the validation set's loss curve showed minor variances, whereas the training set showed a steady decline. These patterns demonstrated the training set's ongoing enhancement and decrease in errors.

The tenth epoch marked the peak for training accuracy and loss, with minor fluctuations observed in validation measurements. Overfitting concerns surfaced since there was a protracted lack of noticeable training improvement. Around the 40th epoch, the accuracy curve saturated, showing diminishing rewards. Training for longer than 40 epochs could cause overfitting. To reduce overfitting and improve model generalization for dependable performance on unknown data, early stopping at the 40th epoch has proven to be essential. With 99.70% training accuracy and 94.57% validation accuracy, the model demonstrated high competency; nevertheless, there was a tiny reduction, which was ascribed to the lesser size of the dataset when compared to the others.

In Fig 13, the presented confusion matrices specifically focus on 7 classes within the Open-Recycle dataset. The model performs admirably when it comes to classifying receipts, with

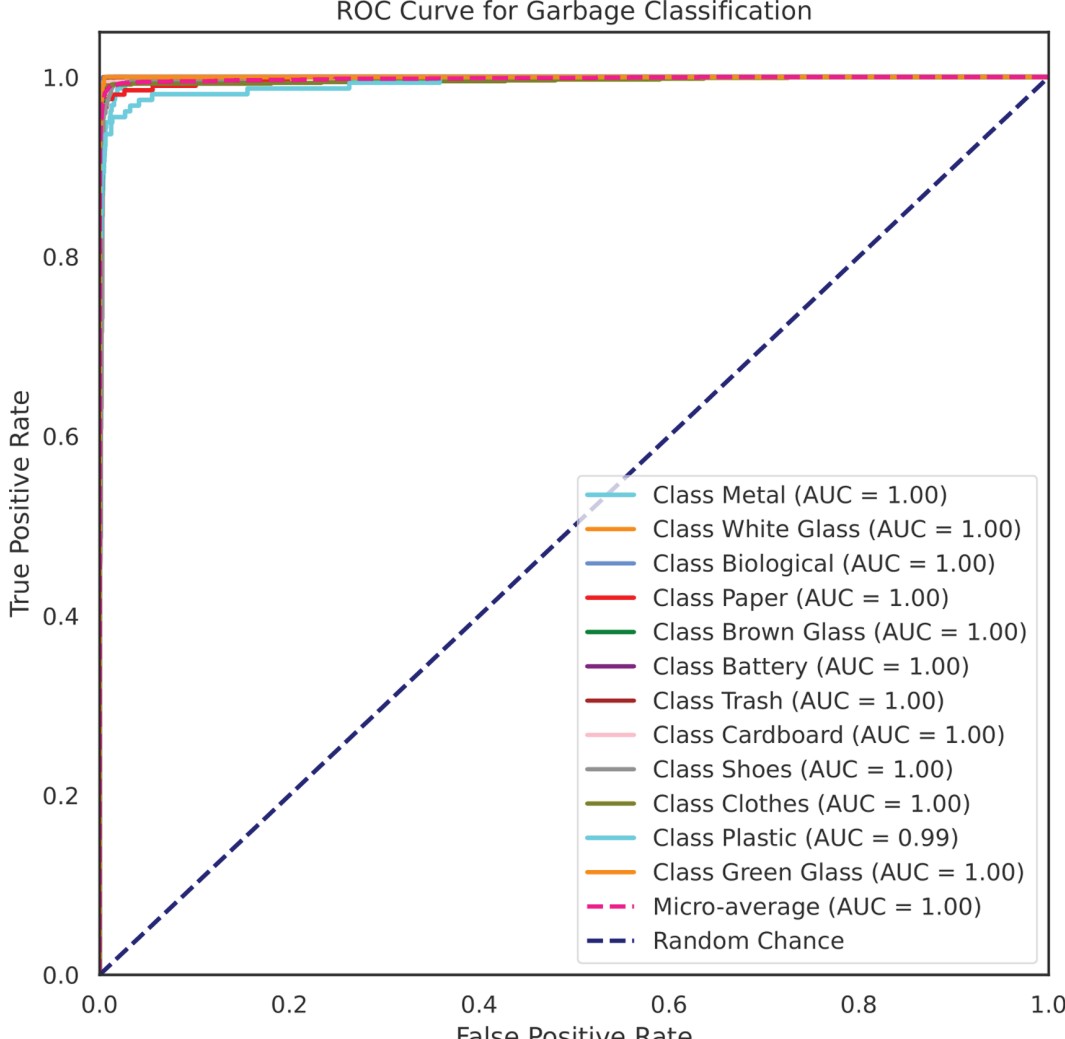

**Fig 11. ROC Curve for Waste Classification dataset.**

never a misclassification, and glass bottles, with only one misclassification. Foil categorization, however, presents difficulties and reveals a comparatively higher percentage of misclassifications. Notably, the model misidentified seven photos of foil as plastic bottles, highlighting a particular area that needs development. These classes show reduced accuracy, which is typified by a higher rate of false negatives and positives. This thorough analysis adds to a nuanced knowledge of the model's performance in the OpenRecycle dataset by illuminating the model's areas of strength and improvement.

Table 6 provides a comprehensive overview of class-wise metrics for the performance evaluation of the model on the OpenRecycle dataset which contains 7 classes. A low percentage of false positives was indicated by precision scores, which regularly ranged from 90.00% to 100.00% and represented the accuracy of positive predictions. Recall scores, which measure how well the model can detect true positives, fared well in the 90.00%–100.00% range, indicating that the majority of instances for each waste class were effectively identified. Exceptionally high F1 scores, ranging from 92.00% to 100.00%, further supported the model's

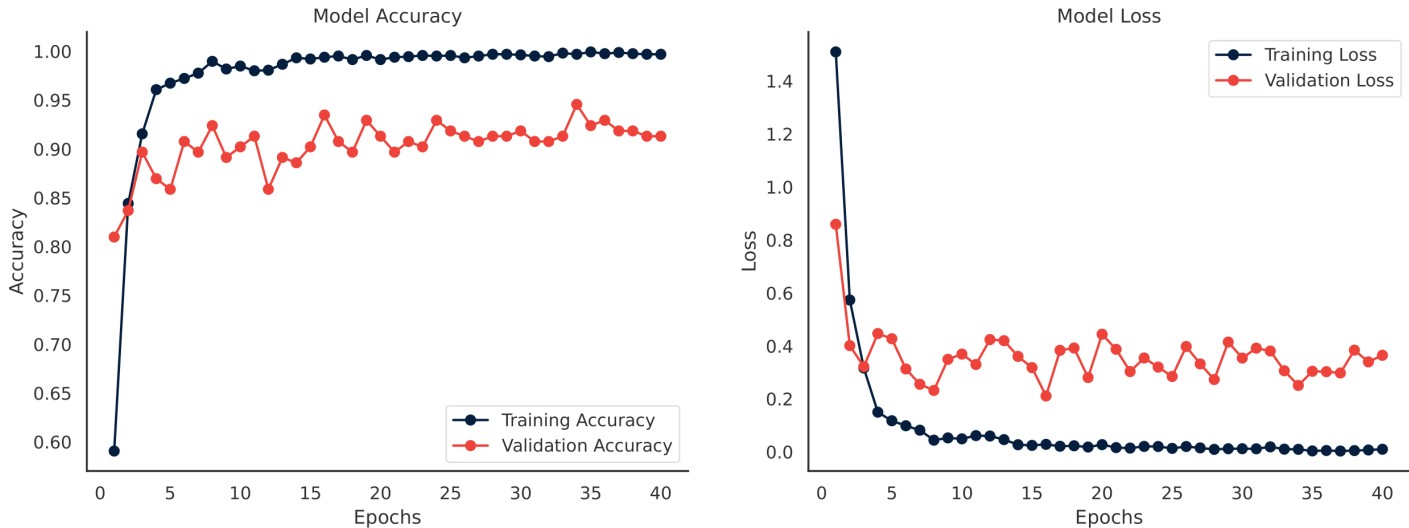

**Fig 12. Accuracy & loss curves of the DenseNet201-Infused Parallel SE-CNN for OpenRecycle Dataset.**

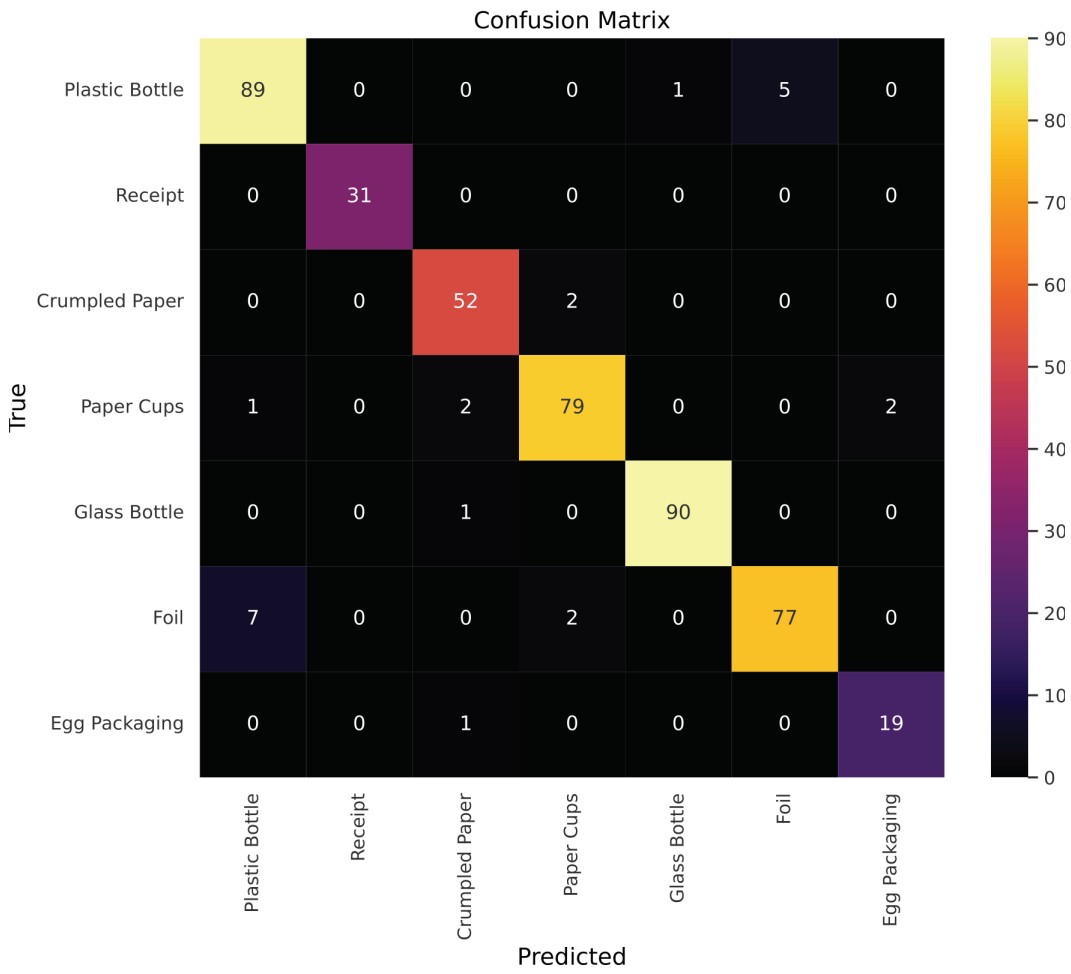

**Fig 13. Confusion Matrix for OpenRecycle Dataset.**

**Table 6. Classwise metrics overview for the OpenRecycle dataset.**

| Class Name | Metrics (%) | | | |
|---|---|---|---|---|
| | Precision | Recall | F1 Score | Specificity |
| Plastic Bottle | 92.00 | 94.00 | 93.00 | 94.00 |
| Receipt | 100 | 100 | 100 | 100 |
| Crumpled Paper | 93.00 | 96.00 | 95.00 | 96.00 |
| Paper Cups | 95.00 | 94.00 | 95.00 | 94.00 |
| Glass Bottle | 99.00 | 99.00 | 99.00 | 99.00 |
| Foil | 94.00 | 90.00 | 92.00 | 90.00 |
| Egg Packaging | 90.00 | 95.00 | 93.00 | 95.00 |

robustness in classification tasks and demonstrated a pleasing trade-off between precision and recall. The metrics indicate a balanced and fair performance across all categories, with no significant discrepancies between classes.

An examination of the metrics in Fig 14 for the OpenRecycle dataset reveals a well-balanced performance. As can be seen in the image, the waste categorization model performs consistently and dependably, with precision, recall, F1 score, specificity, and accuracy all regularly recording at 95.00%. This consistency highlights a successful trade-off between lowering false positives and false negatives, validating the model's robustness in classification tasks. A specificity score of 95.00% further demonstrates the model's capacity to reliably identify true negatives, reducing the possibility of misclassifying other materials. Additionally, the MCC and Kappa scores of 0.94 demonstrate a significant correlation between the predicted and actual classifications, highlighting the model's strength and superior quality. The GDR score of 95.00% further underscores the model's well-rounded performance in effectively recognizing both true positives and true negatives.

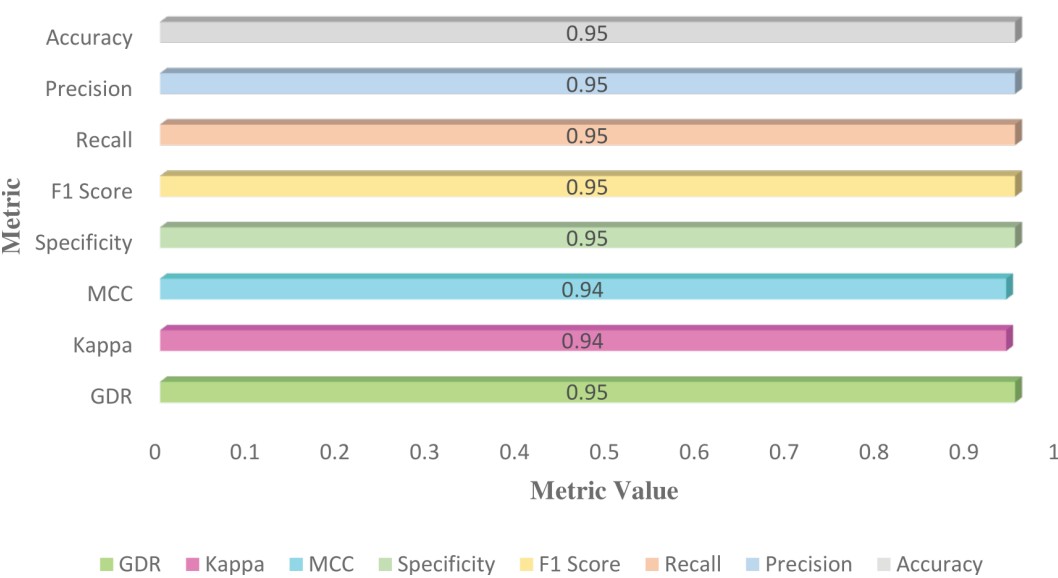

**Fig 14. Overall Metrics overview for the OpenRecycle dataset.**

Fig 15 shows the ROC curve, a graphical depiction of a binary classifier's performance. It displays excellent results for the OpenRecycle dataset, showing an AUC of 1.0 for all classes (foil excluded). The model's exceptional ability to distinguish between positive and negative instances across the majority of categories is indicated by its high AUC. The classifier's great ability to discriminate between positive and negative examples within this particular category is seen in the foil class's AUC score of 0.99. Although the model performs admirably, there are sometimes small categorization errors.

## 5.4 Performance analysis of TrashNet dataset

The accuracy and loss curves in Fig 16 provide a comprehensive overview of the DenseNet201-infused SE integrated Parallel CNN model's performance on the TrashNet dataset. The model initially showed a training accuracy of 59.00% with a bigger training loss, in contrast to the validation set, which attained an accuracy of 68.50% with a lower loss. Backpropagation

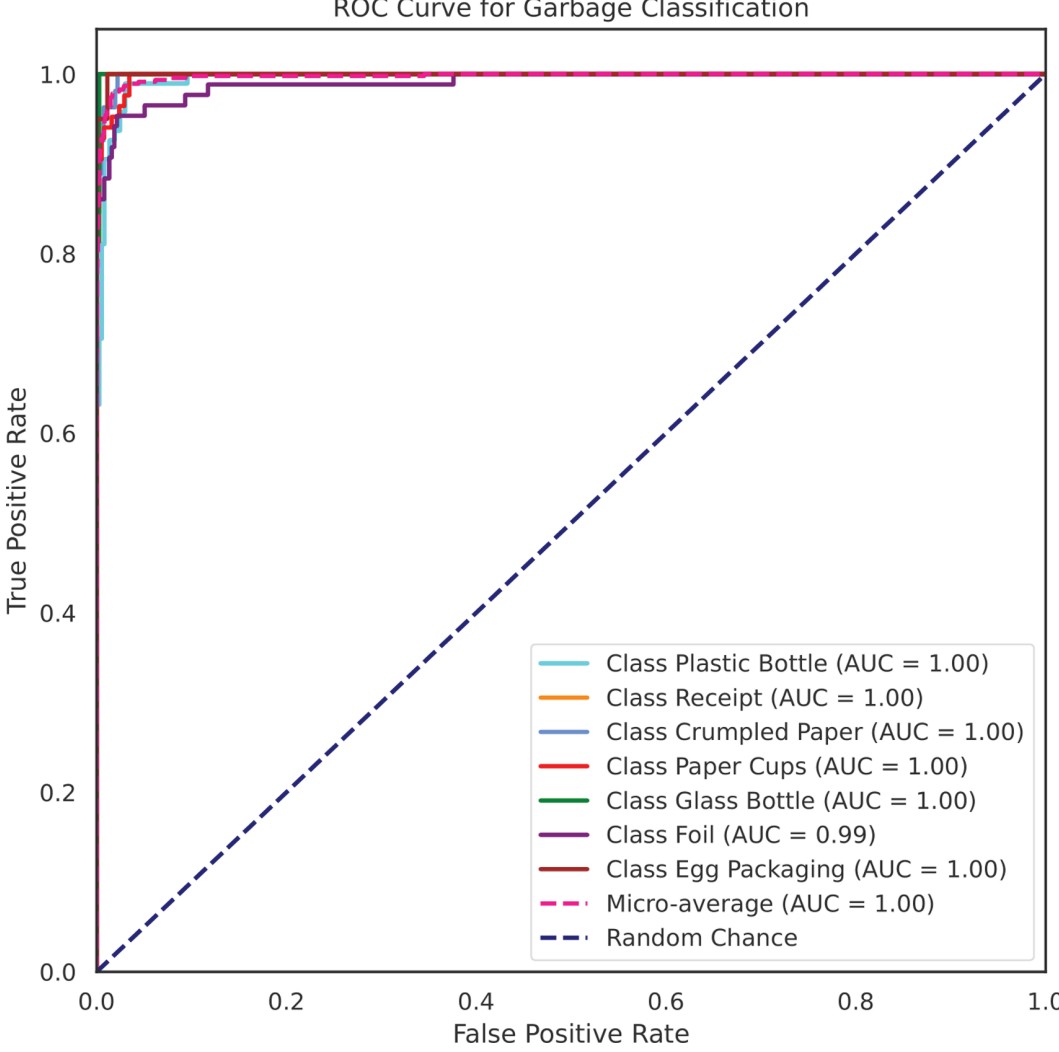

**Fig 15. ROC Curve for OpenRecycle Dataset.**

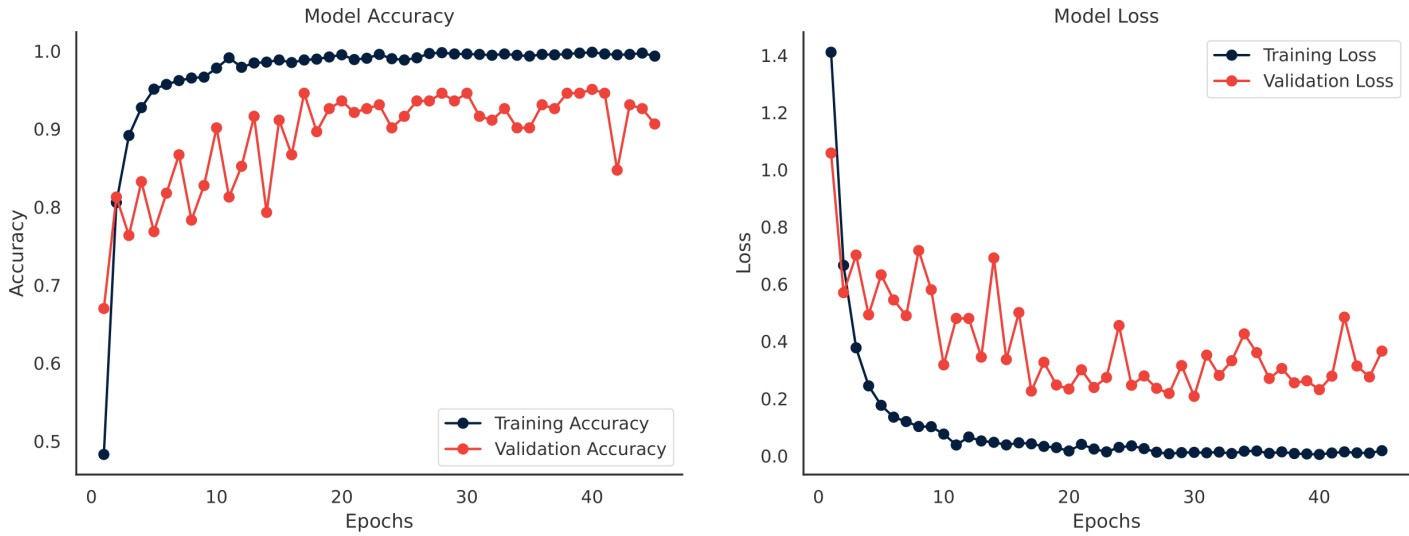

**Fig 16. Accuracy & loss curves of the DenseNet201-Infused Parallel SE-CNN for TrashNet Dataset.**

caused training accuracy and loss to decrease as training went on. The training dynamics in Fig 16 show that there is a critical turning point at approximately 45 epochs, which is marked by an accuracy plateau that indicates diminishing returns to learning. This is consistent with the loss curve stabilizing as the model gets closer to its ideal state. Overfitting could occur if training is carried out for more than 45 epochs.

The model's robust learning is highlighted by the accuracy graph, which shows a considerable ascent to 99.72% over 45 epochs, while validation accuracy plateaus at 94.00% after 18 epochs. Variations in the graph indicate difficulties with model adaptation, which could be related to problems with overfitting or convergence. Nevertheless, it is important to notice that the small size of the dataset leads to larger swings, emphasizing the effect of data scarcity on training stability. The model for this dataset retains an excellent validation accuracy of 95.07% and a remarkable training accuracy of 99.70% despite these difficulties.

In Fig 17, confusion matrices for 6 classes in the Trashnet dataset are illustrated. The model demonstrates strong proficiency in classifying metal, glass, and plastic with minimal misclassifications. However, its performance is comparatively less accurate in classifying trash, as evidenced by a notable number of misclassifications. This lower accuracy for these classes is reflected in an increased occurrence of both false positives and false negatives.

In Table 7, a detailed breakdown of class-wise metrics offers a thorough evaluation of the model's performance on the TrashNet dataset. The precision scores exhibited a continuous range of 89% to 98%, which suggests a low false positive rate and high accuracy of positive predictions. The model's recall scores, excluding the "Trash" class, fell between 93% and 99%, suggesting that the majority of instances for each waste class were effectively identified. The model's resilience in classification tasks was further highlighted by very high F1 scores, ranging from 94% to 97% (without considering the "Trash" class), demonstrating a well-balanced trade-off between recall and precision.

However, the "Trash" class shows a notable dip in performance, with recall at 80.00% and F1 score at 84.00%, indicating the model's reduced effectiveness in identifying true positives for this category. In contrast, the Garbage Classification V2 dataset achieved recall rates of 97% and 99% for the same class. The poor results in the TrashNet dataset stem from the

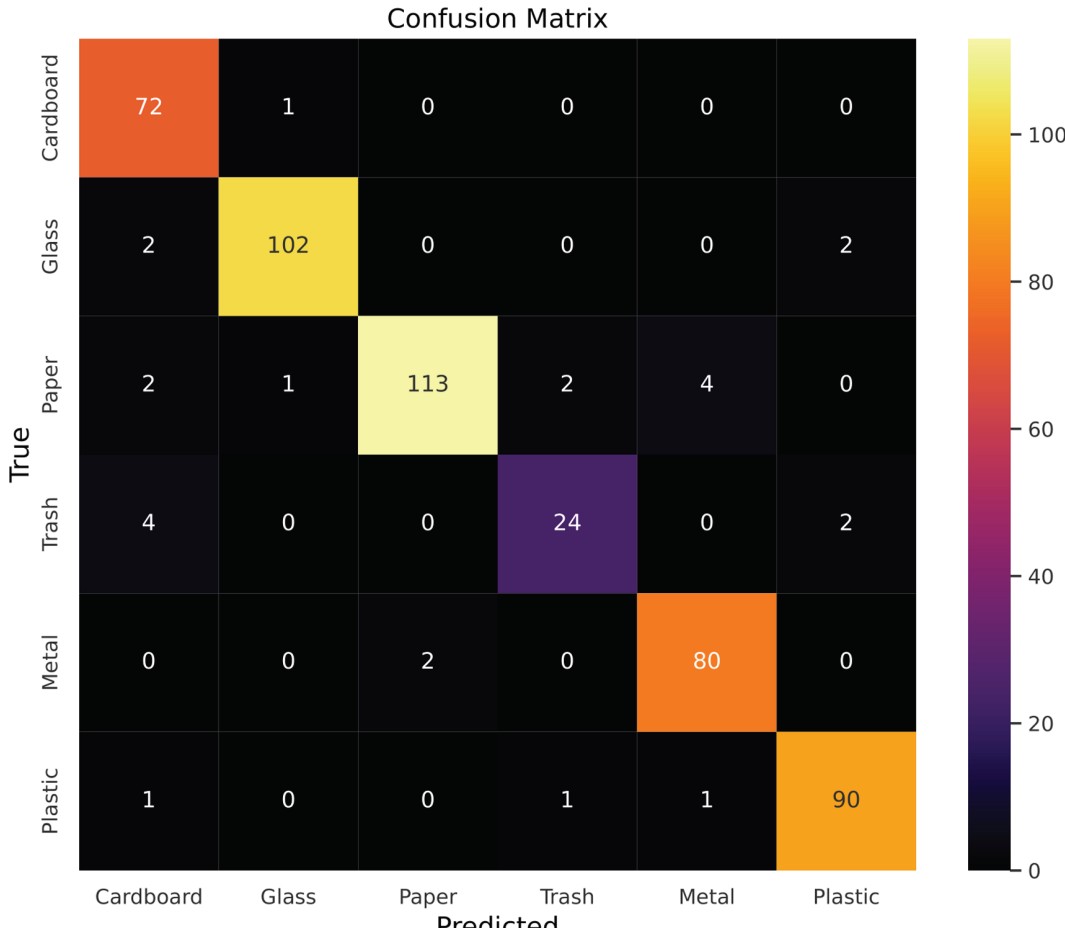

**Fig 17. Confusion Matrix for TrashNet Dataset.**

**Table 7. Classwise metrics overview for the TrashNet dataset.**

| Class Name | Metrics (%) | | | |
|---|---|---|---|---|
| | Precision | Recall | F1 Score | Specificity |
| Cardboard | 89.00 | 99.00 | 94.00 | 99.00 |
| Glass | 98.00 | 96.00 | 97.00 | 96.00 |
| Paper | 98.00 | 93.00 | 95.00 | 93.00 |
| Trash | 89.00 | 80.00 | 84.00 | 80.00 |
| Metal | 94.00 | 98.00 | 96.00 | 98.00 |
| Plastic | 96.00 | 97.00 | 97.00 | 97.00 |

"Trash" class having only 137 images, compared to over 400 images in other classes. This type of data imbalance is a form of dataset bias, as the model becomes predisposed to favor classes with more samples, impacting its ability to identify and classify "Trash" instances accurately. This imbalance can be addressed through data augmentation techniques to increase the image count for the "Trash" class. By employing ImageDataGenerator with rotation, width shift, height shift, shearing, zooming, and horizontal flipping, the image count was balanced across classes. The class-wise metrics after augmentation are shown in Table 8.

**Table 8. Classwise metrics overview after data augmentation for TrashNet dataset.**

| Class Name | Metrics (%) | | | |
|---|---|---|---|---|
| | Precision | Recall | F1 Score | Specificity |
| Cardboard | 99.00 | 96.00 | 97.00 | 96.00 |
| Glass | 92.00 | 100 | 96.00 | 100 |
| Paper | 98.00 | 93.00 | 95.00 | 93.00 |
| Trash | 96.00 | 100 | 98.00 | 100 |
| Metal | 98.00 | 100 | 99.00 | 100 |
| Plastic | 100.00 | 92.00 | 96.00 | 92.00 |

After applying data augmentation techniques, the table clearly shows that the bias problem for the "Trash" class in the TrashNet dataset has been effectively resolved. The "Trash" class now boasts a recall of 100% and an F1 score of 98%, a significant improvement from the previous recall of 80% and F1 score of 84%. Results for the other classes have also improved. However, it's important to note that while data augmentation adds complexity to an already intricate model, it enhanced performance for the underrepresented "Trash" class, which struggled due to significant class imbalance. Although the original approach using just the raw dataset yielded good results, data augmentation can be applied if needed to further mitigate the imbalance issue, potentially achieving even better results with the proposed model.

Analyzing the metrics presented in Fig 18 for the TrashNet dataset reveals a balanced performance by the model. The waste categorization model constantly tracks at 95.00% for precision, recall, F1 score, specificity, and accuracy, indicating its dependability and reliability. The model's ability to maintain a well-balanced recall and precision is highlighted by the outstanding F1 score. A specificity score of 95.00% further highlights the model's ability to consistently

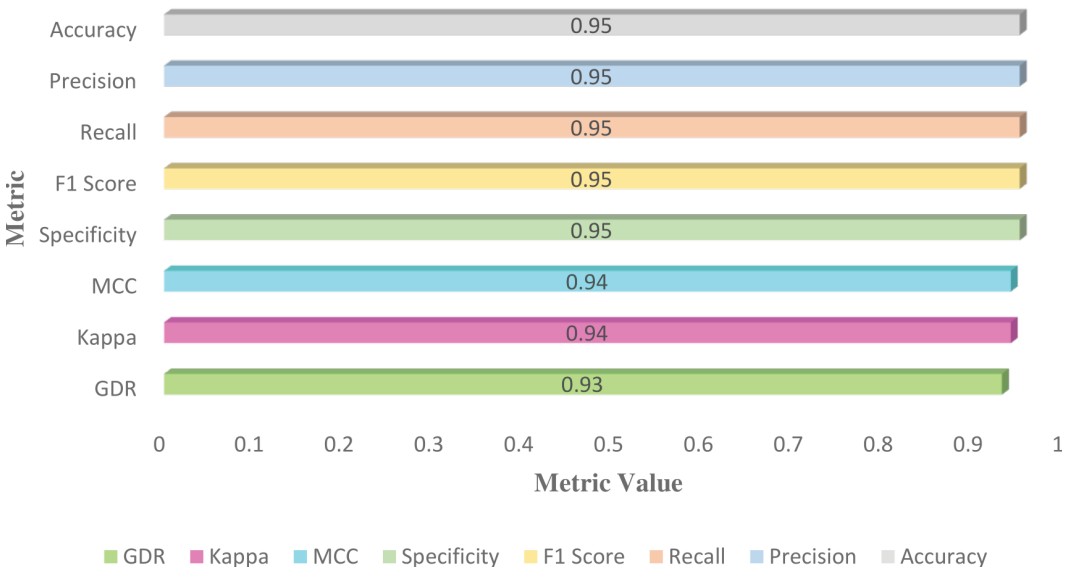

**Fig 18. Overall Metrics overview for TrashNet dataset.**

detect genuine negatives, reducing the possibility of incorrectly classifying other items. Additionally, the MCC and Kappa scores of 0.94 indicate excellent agreement between predicted and actual classifications, reinforcing the model's accuracy and robustness. The GDR score of 93.00% further supports these findings by showcasing the model's effective performance.

In Fig 19, the Receiver Operating Characteristic (ROC) curve provides a comprehensive visualization of the model's performance on the TrashNet dataset. The results showcase an outstanding achievement, as evidenced by the Area Under the Curve (AUC) of 1.0 across all classes except for glass and trash. This highlights the model's exceptional ability to accurately differentiate between positive and negative examples across diverse categories. For the glass and trash class, the AUC score of 0.99 demonstrates the classifier's strong proficiency in distinguishing instances, though minor classification errors occasionally occur. Overall, the consistently high AUC scores affirm the model's effectiveness in handling complex classification tasks.

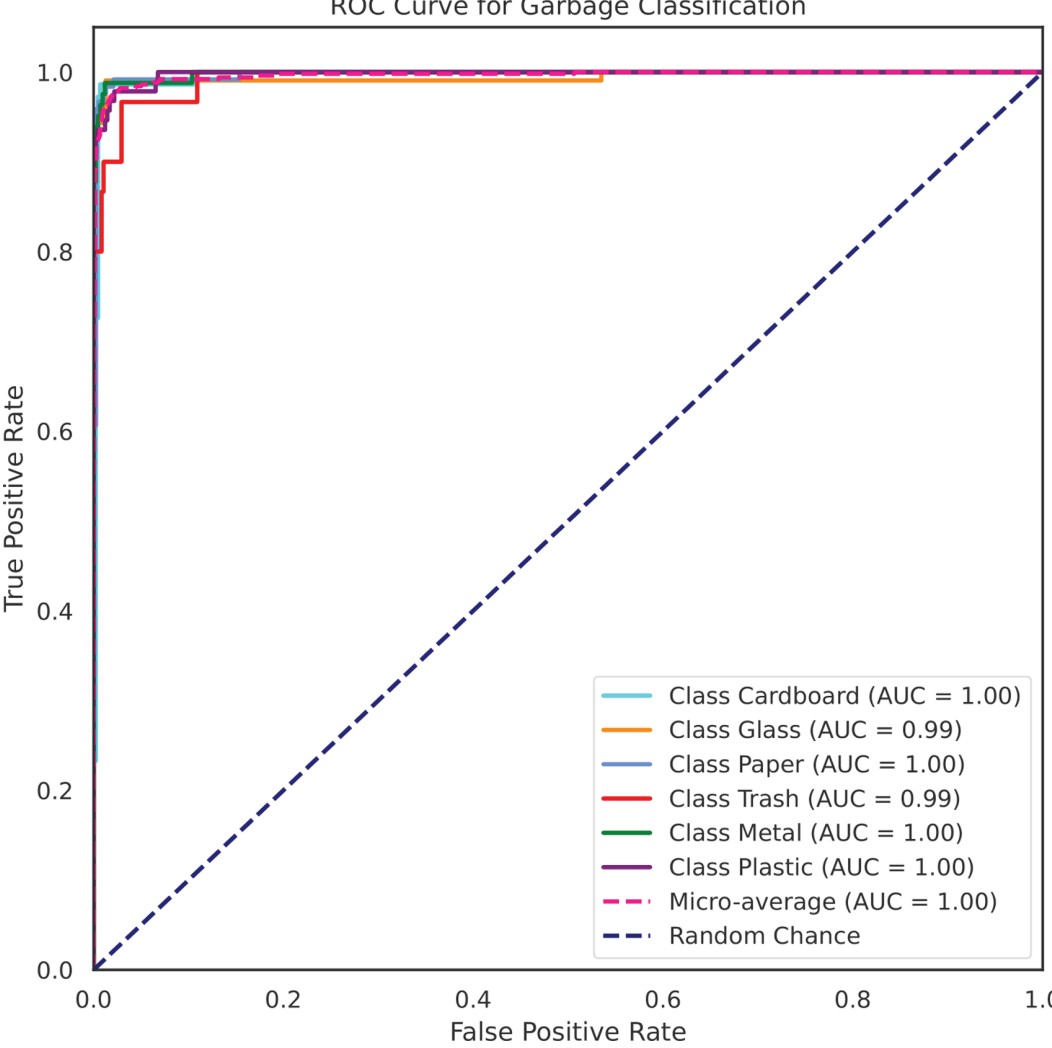

**Fig 19. ROC Curve for TrashNet Dataset.**

## 5.5 Robustness evaluation under adversarial attacks and environmental variability

In Table 9, the robustness evaluation of the model is presented under adversarial attacks, noisy data, and diverse environmental conditions, including variations in lighting and weather. Raw images from the datasets are used for evaluation without preprocessing, ensuring assessment in a real-world scenario. However, rigorously testing the model's performance under additional noise, adversarial attacks, and lighting changes is essential to accurately reflect real-world conditions.

The preprocessing pipeline introduces moderate modifications, adjusting brightness, contrast, and saturation within a range of 0.6 to 1.3, along with minimal hue variations of -5 to +6 degrees, gamma correction for exposure discrepancies, and slight sharpening. These modifications diversify the dataset, improving the model's ability to generalize in various lighting and environmental conditions. Additionally, Gaussian noise with a standard deviation of 25 adds moderate graininess, while salt and pepper noise introduces sporadic white and black pixels at a controlled probability of 10%, creating realistic distortion. The FGSM adversarial attack applies small, targeted perturbations that effectively mislead the model while maintaining plausibility.

The data presented in Table 9 reveals that the accuracy across the four datasets fluctuates under these demanding conditions, with each dataset experiencing significant declines attributed to adversarial effects and noise. For example, the Waste Classification V2 dataset shows a decrease in accuracy of 3.15%, whereas the TrashNet dataset experiences a more pronounced decline of 4.58%. Additionally, under varying environmental conditions, accuracy reductions range from 1.17% for the Waste Classification dataset to 2.54% for the Waste Classification V2 dataset. Notwithstanding these declines, the model exhibits a noteworthy degree of robustness when faced with increased noise, adversarial challenges, and changes in lighting. It successfully retains a substantial portion of its accuracy, demonstrating its ability to address the difficult conditions often encountered in real-world applications. This resilience indicates that the model is well-prepared to manage various disruptions, thereby enhancing its reliability in practical situations.

## 5.6 Validation across additional datasets

In Table 10, the results of the proposed waste classification model, which utilizes a DenseNet201-infused Parallel SE-CNN architecture, are presented using three additional

**Table 9. Robustness evaluation of model under adversarial attacks, noisy data, and environmental conditions.**

| Dataset | Accuracy (%) | Precision (%) | Recall (%) | F1 Score (%) | Accuracy Drop (%) |
|---|---|---|---|---|---|
| **Adversarial Attacks & Noisy Data** | | | | | |
| **Waste Classification V2** | 94.36 | 94.40 | 94.36 | 94.38 | 3.15 |
| **Waste Classification** | 94.71 | 94.83 | 94.71 | 94.75 | 2.01 |
| **OpenRecycle** | 93.49 | 93.59 | 93.49 | 93.54 | 1.37 |
| **TrashNet** | 90.71 | 90.85 | 90.71 | 90.76 | 4.58 |
| **Diverse Environmental Conditions (Lighting, Weather)** | | | | | |
| **Waste Classification V2** | 94.95 | 95.02 | 94.95 | 94.97 | 2.54 |
| **Waste Classification** | 95.52 | 95.58 | 95.52 | 95.55 | 1.17 |
| **OpenRecycle** | 92.41 | 92.48 | 92.41 | 92.43 | 2.51 |
| **TrashNet** | 93.28 | 93.27 | 93.28 | 93.27 | 1.87 |

**Table 10. Performance metrics of the model across three additional datasets.**

| Dataset | Metrics (%) | | | | | MCC | Kappa |
|---|---|---|---|---|---|---|---|
| | Accuracy | Precision | Recall | F1-Score | Specificity | | |
| Waste Images | 94.67 | 94.71 | 94.67 | 94.69 | 94.67 | 0.94 | 0.94 |
| Recyclable Waste Images | 92.77 | 92.91 | 92.77 | 92.83 | 92.77 | 0.92 | 0.92 |
| RealWaste | 93.17 | 93.26 | 93.26 | 93.25 | 93.26 | 0.92 | 0.92 |

datasets that introduce further waste diversity. These datasets—Waste Images, Recyclable Waste Images, and RealWaste—were selected to assess the model's ability to generalize across different types of waste. The model achieves an accuracy of 94.67% on Waste Images, 92.77% on Recyclable Waste Images, and 93.17% on RealWaste, highlighting its strong generalization capabilities. Precision rates of 94.71%, 92.91%, and 93.26% reflect its ability to identify true positives, while recall scores mirror this performance at 94.67%, 92.77%, and 93.26%. F1-scores further reinforce the model's reliability, achieving 94.69%, 92.83%, and 93.25%. Additionally, the Matthews correlation coefficient (MCC) scores 0.94 for Waste Images and 0.92 for the other two datasets, with Kappa values similarly reflecting this trend. The model exhibits stable performance across these three supplementary datasets, reflecting the high accuracy noted in the first four datasets. This consistency highlights its ability to effectively manage different categories of waste.

## 5.7 Performance comparison with different deep CNN architectures and state of the art researches

A comparative examination of the classification performance of the tested models is shown in Table 11. Among the models that were analyzed, the DenseNet201 injected parallel SE-CNN

**Table 11. Comparative analysis of classification performance with different deep CNN models.**

| Evaluated Models (Deep CNN) | Accuracy(%) Waste Classification V2 Dataset | Accuracy(%) Waste Classification Dataset | Accuracy(%) OpenRecycle Dataset | Accuracy(%) TrashNet Dataset | Total Params (Million) |
|---|---|---|---|---|---|
| MobileNet + CA | 96.42 | 95.00 | 90.23 | 90.31 | 12.50 |
| ResNet101 + SA | 96.90 | 95.42 | 89.15 | 91.10 | 57.77 |
| Xception + SA | 97.13 | 96.13 | 91.10 | 88.74 | 35.97 |
| DenseNet169 + CA | 97.06 | 95.97 | 91.10 | 89.52 | 18.91 |
| EfficientNetB5 + Custom Callback | 94.95 | 94.36 | 93.92 | 93.47 | 29.05 |
| DenseNet201 + ELM | 94.24 | 93.42 | 40.56 | 82.51 | 18.32 |
| VGG19 + ELM | 85.14 | 82.82 | 31.67 | 19.96 | 20.02 |
| EfficientNetB7 + RF | 62.41 | 58.01 | 47.29 | 56.13 | 64.10 |
| Ensemble Method + SE | 94.90 | 93.42 | 93.05 | 90.71 | 47.01 |
| InceptionResnetV2 + SE | 96.72 | 92.07 | 94.36 | 87.15 | 70.47 |
| DenseNet201 + CA | 94.97 | 94.94 | 93.05 | 90.71 | 24.61 |
| DenseNet201 + SA | 93.20 | 92.89 | 92.84 | 90.11 | 27.27 |
| MobileNetV2 + parallel SE-CNN | 93.97 | 95.29 | 91.75 | 88.54 | 26.69 |
| DenseNet121 + parallel SE-CNN | 96.41 | 96.19 | 94.57 | 92.09 | 28.26 |
| MobileNet + parallel SE-CNN | 95.26 | 95.32 | 93.27 | 91.11 | 24.46 |
| NASNetMobile + parallel SE-CNN | 95.36 | 95.90 | 91.97 | 89.72 | 25.90 |
| **Proposed Method** | **97.43** | **96.65** | **94.79** | **95.06** | **50.79** |

model is the best option since it consistently shows higher and more stable accuracy on all tested datasets. The suggested strategy beats its equivalents in every case, with accuracy scores of 97.43%, 96.65%, 94.79%, and 95.06% on the Waste Classification v2 Dataset, Waste Classification Dataset, OpenRecycle Dataset, and TrashNet Dataset, respectively. The suggested model consistently outperforms other models, like DenseNet169 with Channel Attention and Xception with Soft Attention, by achieving greater accuracy rates on all the datasets. The performance of these alternative models drops to about 90.00% when applied to the OpenRecycle and TrashNet datasets, which have minimal or short data, even if they display competitive accuracy levels for the first two datasets, Waste Classification v2 and Waste Classification. Conversely, the suggested model exhibits exceptional robustness and flexibility, preserving higher accuracy throughout the range of evaluated datasets.

In the comparative analysis, three models stand out for their high accuracy and low complexity. MobileNet with channel attention achieved impressive accuracies of 96.42% on the Waste Classification V2 dataset, 95.00% on the Waste Classification dataset, 90.23% on the OpenRecycle dataset, and 90.31% on the TrashNet dataset, with a low parameter count of just 12.50 million. DenseNet201 with channel attention closely followed, showing consistent performance with accuracies of 94.97%, 94.94%, 93.05%, and 90.71%, requiring 24.61 million parameters. DenseNet121 + parallel SE-CNN also performed well, achieving accuracies of 96.41%, 96.19%, 94.57%, and 92.09% with 28.26 million parameters. Other TL models tested with the parallel SE-CNN architecture did not match the performance of DenseNet201 in this configuration. Nonetheless, all were consistently outperformed by the proposed model with approximately 50 million parameters. While the higher parameter count indicates increased complexity, it is essential for achieving optimal and reliable results. However, in resource-constrained settings, other models mentioned with lower parameters—approximately one-third to one-half of the proposed model—can be more suitable, as they still achieve over 90% accuracy across all four datasets.

The success of the DenseNet201-infused parallel CNN with Squeeze-and-Excitation (SE) lies in its efficient information flow through dense connectivity, allowing the model to capture intricate patterns. The resilience of the model is increased by the parallel CNN design, which processes data through multiple paths at once to improve feature extraction. By dynamically modifying channel priority, highlighting important characteristics, and suppressing less useful ones, the SE technique improves feature representation. The model's ability to recognize subtle patterns is enhanced by this synergy, which makes it a strong option for intricate image categorization tasks.

The comparison presented in Table 12 highlights the performance of various machine learning classifiers applied to features extracted from the proposed custom architecture, as

**Table 12. ML classifiers performance comparison across datasets.**

| Machine Learning Classifiers | Accuracy(%) Waste Classification V2 Dataset | Accuracy(%) Waste Classification Dataset | Accuracy(%) OpenRecycle Dataset | Accuracy(%) TrashNet Dataset |
|---|---|---|---|---|
| KNN | 83.93 | 80.66 | 75.27 | 67.00 |
| SVM | 85.09 | 84.11 | 80.26 | 69.76 |
| Random Forest | 72.69 | 66.36 | 72.67 | 60.08 |
| Decision Tree | 49.33 | 44.22 | 51.19 | 38.34 |
| Logistic Regression | 82.48 | 81.70 | 80.26 | 68.38 |
| Naive Bayes | 59.89 | 62.87 | 70.50 | 44.27 |
| AdaBoost | 56.88 | 57.14 | 39.26 | 45.06 |
| **Proposed Method** | **97.43** | **96.65** | **94.79** | **95.06** |

opposed to the fully connected layer used in the deep CNN models. Among the classifiers evaluated, k-Nearest Neighbors (KNN), Support Vector Machine (SVM), and logistic regression demonstrate comparatively higher accuracy than other classifiers. Despite achieving reasonable accuracy, these machine learning classifiers are limited to approximately 80% accuracy for larger datasets and drop below 70% for the TrashNet dataset. This performance is notably inferior to the accuracy levels attained by the proposed method. The suggested architecture, which employs a fully connected layer along with multiple dense layers, yields accuracy scores that significantly surpass those of conventional classifiers. The use of fully connected layers enhances results by effectively learning complex, non-linear relationships and synthesizing features into a more advanced representation. This improved feature learning enables the model to identify intricate patterns and achieve more precise classifications, thereby outperforming traditional machine learning classifiers.

A Comparative Analysis of classification performance among models in previous studies is shown in Table 13. A fascinating trend is that models are mostly tested on a single dataset, which raises legitimate questions regarding their wider applicability to different data sets. On the other hand, the proposed model stands out due to its comprehensive evaluation across four distinct datasets, supplemented by assessments on three additional datasets. This thorough approach not only complements the findings from the initial four datasets but also provides a nuanced understanding of the model's performance in categorizing waste materials, given the diverse sizes and classes of all seven datasets. This comprehensive analysis highlights the flexibility of the suggested DenseNet201-powered parallel CNN with a

**Table 13. Comparative analysis of classification performance among models from prior studies.**

| Ref. No. | Dataset Used | No. of Classes | Model | Accuracy (%) | Precision (%) | Recall (%) | F1 score(%) |
|---|---|---|---|---|---|---|---|
| [10] | garbage classification | 6 | ResNet50 | 95.35 | - | - | - |
| [13] | Custom dataset | 4 | VGG16 | 75.60 | - | - | - |
| [14] | TrashNet | 6 | MobileNet | 87.20 | - | - | - |
| [16] | TrashNet | 6 | Xception | 88.00 | 83.00 | 84.00 | - |
| [17] | Custom dataset | 5 | GAF_Dense | 92.10 | - | - | - |
| [19] | Custom dataset | 4 | InceptionV3 | 93.20 | - | - | - |
| [21] | TrashNet | 6 | MLH-CNN | 92.60 | - | - | - |
| [22] | TrashNet | 6 | MobileNetV2 + SVM | 94.28 | 94.70 | 93.10 | - |
| [23] | Custom dataset | 4 | Modified Algorithm | 95.00 | - | - | - |
| [24] | Baidu dataset | 4 | MobileNetV3 + LSTM | 81.00 | - | - | - |
| [25] | Custom dataset | 7 | EfficientNet-B2 | 75.00 | - | - | - |
| [26] | TrashNet | 6 | DenseNet121 | 95.00 | - | - | - |
| [27] | Custom dataset | 6 | CNN | 92.50 | - | - | - |
| [32] | Waste Images | 9 | Ensemble | 90.00 | - | - | - |
| [35] | RealWaste | 9 | Inception V3 | 89.19 | 91.34 | 87.73 | 90.25 |
| [37] | TrashNet + Google Images | 7 | Ensemble | 94.11 | - | - | - |
| [38] | TrashNet | 6 | Densenet169 | 94.90 | 94.50 | 92.50 | - |
| [39] | garbage classification | 6 | ResNet101 | 87.76 | 85.70 | 87.94 | 86.81 |
| **Our Work** | **Waste classification v2** | **10** | **Proposed Method** | **97.43** | **97.45** | **97.43** | **97.43** |
| | **Waste classification** | **12** | | **96.65** | **96.66** | **96.63** | **96.64** |
| | **OpenRecycle** | **7** | | **94.79** | **94.81** | **94.79** | **94.77** |
| | **TrashNet** | **6** | | **95.06** | **95.09** | **95.05** | **95.06** |
| | **Waste Images** | **9** | | **94.67** | **94.71** | **94.67** | **94.69** |
| | **Recyclable Waste Images** | **9** | | **92.77** | **92.91** | **92.77** | **92.83** |
| | **RealWaste** | **9** | | **93.17** | **93.26** | **93.26** | **93.25** |

squeeze-and-excitation mechanism and provides important details on the subtleties of its performance in different data distributions.

Historically, common approaches have been strongly biased toward using the TrashNet dataset or custom datasets created by merging or web crawling. The disadvantage, however, was the absence of thorough validation over a wide range of datasets, which left a crucial knowledge vacuum regarding the models' flexibility outside of certain training scenarios. On the other hand, the suggested model has proven to be a strong solution, exhibiting outstanding results on datasets with different sizes and class distributions. It is noteworthy because not only did it outperform its predecessors in tests that were explicitly focused on the TrashNet dataset, but it also showed exceptional performance on datasets with different levels of complexity. This thorough assessment validates the proposed model's adaptability and highlights its potential for successful implementation across datasets with different features, resolving a significant shortcoming in earlier approaches.

Furthermore, comparing the proposed model to counterparts in previous research, the results shown in Table 13 show that it consistently achieves superior accuracy, ranging from 92.77% to 97.43%. This ongoing superiority demonstrates the method's dependability and efficacy, which are essential for a classification model's actual applicability. In addition to accuracy, the table's inclusion of precision, recall, and F1 score measures highlights the model's careful examination and reliable performance assessment.

To evaluate the individual contributions of the parallel CNN branches and the SE attention mechanism, an ablation study was conducted to further validate why the proposed model is indeed superior. Four variants were tested, excluding the proposed method, to dissect the impact of each component and understand the synergistic effects when combined. As shown in Table 14, the performance of each configuration was assessed across four datasets, using accuracy, precision, recall, and F1-score as key evaluation metrics.

The DenseNet201-only model established a reliable baseline, with accuracy ranging from 92.49% to 95.48%. However, despite being consistent, it lacked further performance gain, suggesting limited adaptability. Adding a custom CNN branch without SE attention or parallel structure (DenseNet201 + CCNN (no SE, no parallel)) led to only marginal improvement in some datasets like "Waste Classification" (96.06%) and even slight regressions in others such as "OpenRecycle (93.05%). When only SE attention was introduced (DenseNet201 + SE-CNN (no parallel)), better refinement was observed—especially on "Waste Classification" and "OpenRecycle" (93.05%)—owing to enhanced channel-wise feature emphasis. On the other hand, the inclusion of parallel CNN branches without SE (DenseNet201 + Parallel CNN (no SE)) demonstrated noticeable variability: "Waste Classification" jumped to 96.51%, yet performance on "Waste Classification V2" declined to 93.16%, revealing that although parallelism aids multi-scale feature learning, it may also amplify noise if unfiltered.

The proposed model, which fuses both the SE attention mechanism and parallel CNN branches into the DenseNet201 backbone, clearly outperforms all previous setups. It achieves the highest accuracy (97.43% on "Waste Classification V2") and maintains a stable and improved score across all datasets, including 96.65% on "Waste Classification" and 95.06% on "TrashNet". This fusion architecture effectively capitalizes on the complementary strengths of both attention and parallelism—allowing for both refined channel weighting and rich hierarchical feature extraction. In conclusion, this ablation analysis underscores that while SE attention and parallel CNNs each contribute value independently, their integration within the proposed model yields a far more generalizable, stable, and high-performing solution.

These thorough measurements highlight the model's capacity to balance reducing false positives and false negatives, adding to a more nuanced knowledge of its capabilities. The

**Table 14. Impact of parallel CNN Branches and SE attention on model performance.**

| Dataset | Model Description | Accuracy (%) | Precision (%) | Recall (%) | F1-score (%) |
|---------|------------------|-------------|--------------|-----------|--------------|
| [28] | DenseNet201 Only | 95.48 | 95.45 | 95.46 | 95.46 |
| [29] | | 95.24 | 95.21 | 95.25 | 95.23 |
| [30] | | 93.57 | 93.58 | 93.57 | 93.57 |
| [31] | | 92.49 | 92.44 | 92.46 | 92.46 |
| [28] | DenseNet201 + CCNN (no SE, no parallel) | 95.19 | 95.19 | 95.20 | 95.19 |
| [29] | | 96.06 | 96.09 | 96.05 | 96.07 |
| [30] | | 93.05 | 93.04 | 93.05 | 93.04 |
| [31] | | 93.47 | 93.47 | 93.44 | 93.45 |
| [28] | DenseNet201 + SE-CNN (no parallel) | 95.44 | 95.49 | 95.44 | 95.45 |
| [29] | | 96.22 | 96.18 | 96.22 | 96.21 |
| [30] | | 94.14 | 94.14 | 94.17 | 94.14 |
| [31] | | 93.28 | 93.22 | 93.25 | 93.24 |
| [28] | DenseNet201 + Parallel CNN (no SE) | 93.16 | 93.26 | 93.17 | 93.13 |
| [29] | | 96.51 | 96.56 | 96.48 | 96.50 |
| [30] | | 93.27 | 93.27 | 93.25 | 93.27 |
| [31] | | 92.49 | 92.41 | 92.53 | 92.45 |
| [28] | **Proposed Method** | **97.43** | **97.45** | **97.43** | **97.43** |
| [29] | | **96.65** | **96.66** | **96.63** | **96.64** |
| [30] | | **94.79** | **94.81** | **94.79** | **94.77** |
| [31] | | **95.06** | **95.09** | **95.05** | **95.06** |

model's constant outperformance across a range of criteria not only demonstrates its versatility and real-world potential but also proves its correctness and establishes it as a dependable option for practical deployment.

## 5.8 Brief overview of empirical time complexity

Table 15 provides a detailed analysis of time complexity across four datasets, emphasizing key metrics like total training time, epoch duration, and average inference time per sample. Inference time is the duration a trained model takes to predict on new data. For the test set, it reflects how quickly the model generates predictions per sample, indicating real-world performance. Training times varied significantly due to dataset size and epoch requirements, with larger datasets such as "Waste Classification V2" having longer epoch durations (210 seconds on average) compared to "OpenRecycle" (37 seconds). This suggests that the computational demands of the model increase in relation to the size of the dataset. Nevertheless, the efficiency of inference remained consistently high, achieving an average rate of 73 ms per step for "Waste Classification V2," which translates to an inference time of 2.28 ms per image. The marginally extended inference times observed for smaller datasets, such as "OpenRecycle," were linked to the larger dimensions of the images, which were selected to eliminate the need for further preprocessing, thus preserving overall efficiency.

**Table 15. Empirical time complexity analysis across datasets.**

| Dataset | Train Time (s) | Epoch Time (s) | | | Epochs | Inference Time | | Images |
|---------|---------------|-----|-----|-----|--------|----------------|---------------|--------|
| | | Min | Max | Avg | | Avg (ms/step) | Avg/Sample (ms) | |
| Waste Classification V2 | 6934.74 | 208 | 213 | 210 | 33 | 73 | 2.28 | 21,983 |
| Waste Classification | 5936.20 | 145 | 148 | 147 | 40 | 76 | 2.38 | 15,515 |
| OpenRecycle | 1523.07 | 36 | 38 | 37 | 40 | 143 | 4.47 | 2,301 |
| TrashNet | 1910.05 | 41 | 42 | 41.5 | 44 | 150 | 4.69 | 2,527 |

The analysis validates the model's exceptional scalability, revealing a nearly linear correlation between the quantity of images and the overall training duration, indicating its capability to handle larger datasets without imposing significant computational burdens. Furthermore, the model demonstrated consistent epoch lengths across different datasets, showcasing its robustness against variations in dataset size and structure. Its quick inference times are essential for real-time applications, ensuring minimal processing delays even as the datasets expand. The model's reliable performance across a range of datasets with differing class counts underscores its flexibility and durability, reducing the necessity for extensive retraining.

## 5.9 Energy consumption assessment

This research involved training and evaluating the model on Kaggle with the aid of a Tesla T4 GPU. The "pynvml" library was employed to monitor GPU power usage, while the "codecarbon" library was utilized to measure carbon emissions. Power consumption was assessed for each training epoch, and energy consumption was determined using the equation Energy = Power×Time, which enabled the total energy consumption to be calculated in joules throughout the training process. After the training concluded, the overall carbon emissions were documented, providing critical insights into the environmental ramifications of the model training procedure.

The results summarized in Table 16 reveal a range of energy consumption and carbon emissions across the analyzed datasets, with the Waste Classification V2 dataset recording the highest energy use at 1593.70 Joules and emissions of 0.0152 kg CO2e, while the OpenRecycle dataset showed the lowest consumption at 1520.27 Joules and minimal emissions of 0.0045 kg CO2e. The TrashNet dataset had significant energy consumption at 1586.57 Joules, resulting in the highest carbon emissions of 0.0191 kg CO2e. These differences arise from factors such as dataset size, training duration, and image dimensions. The model demonstrates varying energy efficiency across datasets, with most exhibiting moderate energy consumption and carbon emissions. Its ability to achieve robust results with relatively low energy use highlights its potential for energy efficiency, which is crucial in deep learning. Considering the inherent computational demands, these results are acceptable within the deep learning context, where performance often requires resource trade-offs. This underscores the need for continuous optimization to enhance energy efficiency while preserving predictive accuracy, ensuring the sustainability of deep learning methodologies as they advance.

## 5.10 Brief benchmarking of model complexity and deployment viability

In Table 17, a comparative benchmarking of five deep learning models—Xception+SA, ResNet101+SA, DenseNet121+Parallel SE-CNN, MobileNet+CA, and the Proposed Model—has been conducted. These architectures were selected based on their high average

**Table 16. Energy consumption and carbon emissions for different datasets.**

| Dataset | Total Energy Consumption (Joules) | Carbon Emissions (kg CO2e) |
|---|---|---|
| Waste Classification V2 | 1593.70 | 0.0152 |
| Waste Classification | 1537.54 | 0.0123 |
| OpenRecycle | 1520.27 | 0.0045 |
| TrashNet | 1586.57 | 0.0191 |
| RealWaste | 1538.44 | 0.0070 |

accuracies as shown in Table 11, where the proposed method achieved 96.23%, followed by DenseNet121+SE-CNN (94.82%), Xception+SA (93.28%), ResNet101+SA (93.14%), and MobileNet+CA (92.99%). To ensure consistency in evaluation, all models were tested using the Waste Classification dataset—selected for its diversity with 12 distinct classes, making it a robust benchmark for deployment viability. All benchmarking experiments were conducted on the Kaggle platform utilizing an NVIDIA Tesla T4 GPU to ensure consistency and reliability in computational performance measurements.

From an empirical standpoint, the Proposed Model clearly dominates in terms of accuracy and reliability, justifying its suitability for deployment in scenarios where performance is critical and computational resources are abundant. However, this performance comes at the cost of increased model complexity—with 50.79M parameters, 5936.20 seconds of training time, and a sizable model footprint of 581.8 MB. The peak memory usage is also highest, reinforcing the model's resource-intensive nature. Consequently, in resource-constrained environments, such as edge devices or mobile deployments, this level of complexity may become a bottleneck.

In such scenarios, alternatives like MobileNet+CA, with its lightweight architecture (12.5M params) and fastest inference time (1.38 ms/sample), emerge as viable substitutes—especially when computational efficiency is prioritized over peak accuracy. Although its average accuracy (92.99%) is the lowest among the selected group, the trade-off may be acceptable for real-time or embedded applications. Meanwhile, DenseNet121+Parallel SE-CNN offers a compelling balance between accuracy (94.82%) and moderate resource requirements, making it a solid middle-ground candidate. So, while the Proposed Model remains the ideal choice for high-reliability systems, practitioners aiming for deployment in low-power or memory-limited settings may consider DenseNet121+ parallel SE-CNN or MobileNet+CA, depending on the acceptable trade-offs between accuracy and efficiency.

## 5.11 Model explainability via grad-CAM

To gain deeper insights into the decision-making process of the proposed DenseNet201-infused parallel SE-CNN model, Gradient-weighted Class Activation Mapping (Grad-CAM) was employed as an explainability tool. Grad-CAM is a widely adopted post hoc interpretability technique designed for convolutional neural networks, which highlights the regions of an input image that significantly influence the model's prediction. Given a specific class $c$, Grad-CAM generates a coarse localization map $L^{\text{Grad-CAM}} \in \mathbb{R}^{u \times v}$ using the gradients of the class score $y^c$ with respect to the feature maps $A^k$ of a convolutional layer. The localization map is

**Table 17. Empirical benchmarking of model complexity and computational requirements across architectures.**

| Model | Params (M) | Inference Time (Avg/Sample, ms) | Total Energy Consumption (J) | Total Training Time (s) | Model Size (MB) | Peak Memory Usage (MB) |
|---|---|---|---|---|---|---|
| Xception + SA | 35.97 | 2.96 | 1361.07 | 3237.13 | 411.9 | 3759.15 |
| ResNet101 + SA | 57.77 | 3.63 | 1564.38 | 5829.53 | 661.7 | 3841.55 |
| DenseNet121 + parallel SE-CNN | 28.26 | 4.09 | 1137.15 | 3468.86 | 324.4 | 3833.83 |
| MobileNet + CA | 12.50 | 1.38 | 1221.71 | 1527.93 | 143.5 | 3746.57 |
| Proposed Model | 50.79 | 2.38 | 1537.54 | 5936.20 | 581.8 | 3913.99 |

computed as:

$$L^{\text{Grad-CAM}} = \text{ReLU}\left(\sum_k \alpha_k^c A^k\right), \quad \text{where} \quad \alpha_k^c = \frac{1}{Z}\sum_i \sum_j \frac{\partial y^c}{\partial A_{ij}^k} \tag{15}$$

Here, $\alpha_k^c$ represents the importance weight for feature map $k$, and $Z$ is the total number of spatial locations in the feature map. This technique thus produces a visual explanation indicating which parts of the input image contributed most to the prediction decision [40].

In the context of the proposed parallel SE-CNN model, which comprises two distinct branches—one incorporating average pooling and the other max pooling—Grad-CAM heatmaps were generated for the last convolutional layer in each branch independently. This approach enabled a comparative analysis of the focus regions across both feature extraction paths. As illustrated in Fig 20(a), the average pooling branch failed to localize any distinct region, whereas the max pooling branch concentrated on semantically relevant areas of the image, resulting in a correct classification. This observation underscores a key benefit of the dual-branch architecture: when one branch lacks discriminatory focus, the complementary branch may successfully compensate by capturing essential features, enhancing overall robustness.

Furthermore, Fig 20(c), 20(f), and 20(h) demonstrate scenarios where both branches effectively attended to the appropriate regions of interest, leading to highly accurate cumulative heatmaps and correct predictions. These examples validate the synergy between the parallel branches in reinforcing relevant spatial information during inference. Conversely, Fig 20(b) highlights a case where the average pooling branch produced no activations, and the max pooling branch fixated on an irrelevant fragment of the input, ultimately yielding a misclassification. A similar failure mode is observed in Fig 20(e), where, despite some effort by the average pooling branch to identify key patterns, the max pooling branch failed to focus on the clothing area, mistakenly identifying the item as paper. These cases reveal that although the ensemble of branches generally offers resilience, the absence of meaningful focus in both can still lead to errors.

Overall, Grad-CAM provided a valuable interpretative lens into the internal mechanics of the parallel SE-CNN framework. It illustrated how the dual-path strategy diversifies feature representation, offering a form of internal redundancy that mitigates the risk of poor localization in single-path networks.

## 6 Real-time deployment of waste classification system

### 6.1 Website deployment

A cutting-edge website that masterfully combines the powers of deep learning models saved as H5 files stands out as a beacon of innovation in the ever-changing field of waste management and environmental sustainability. The platform has been painstakingly designed to provide a visually appealing and straightforward user experience. This is achieved through the seamless integration of HTML and CSS for appealing aesthetics. In addition, a script.js file skillfully integrates Flask functionality into the website's design. The website makes use of a trained deep learning model that is saved in HDF5 format and integrated into the back-end using Python Flask as the primary framework. Through the easy-to-use upload feature for wasted objects, the platform makes use of the quick processing speed of these

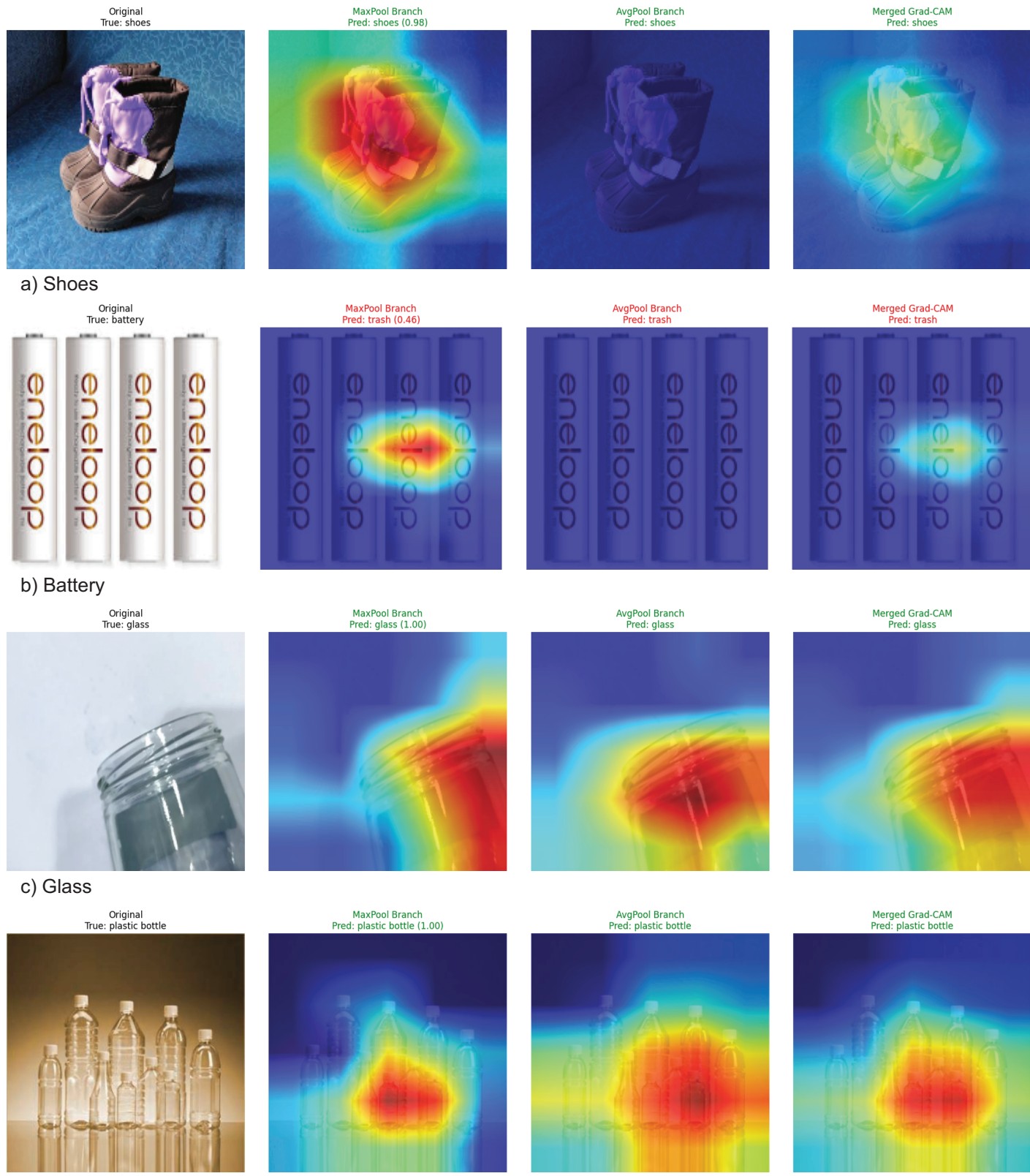

a) Shoes

b) Battery

c) Glass

d) Plastic Bottle

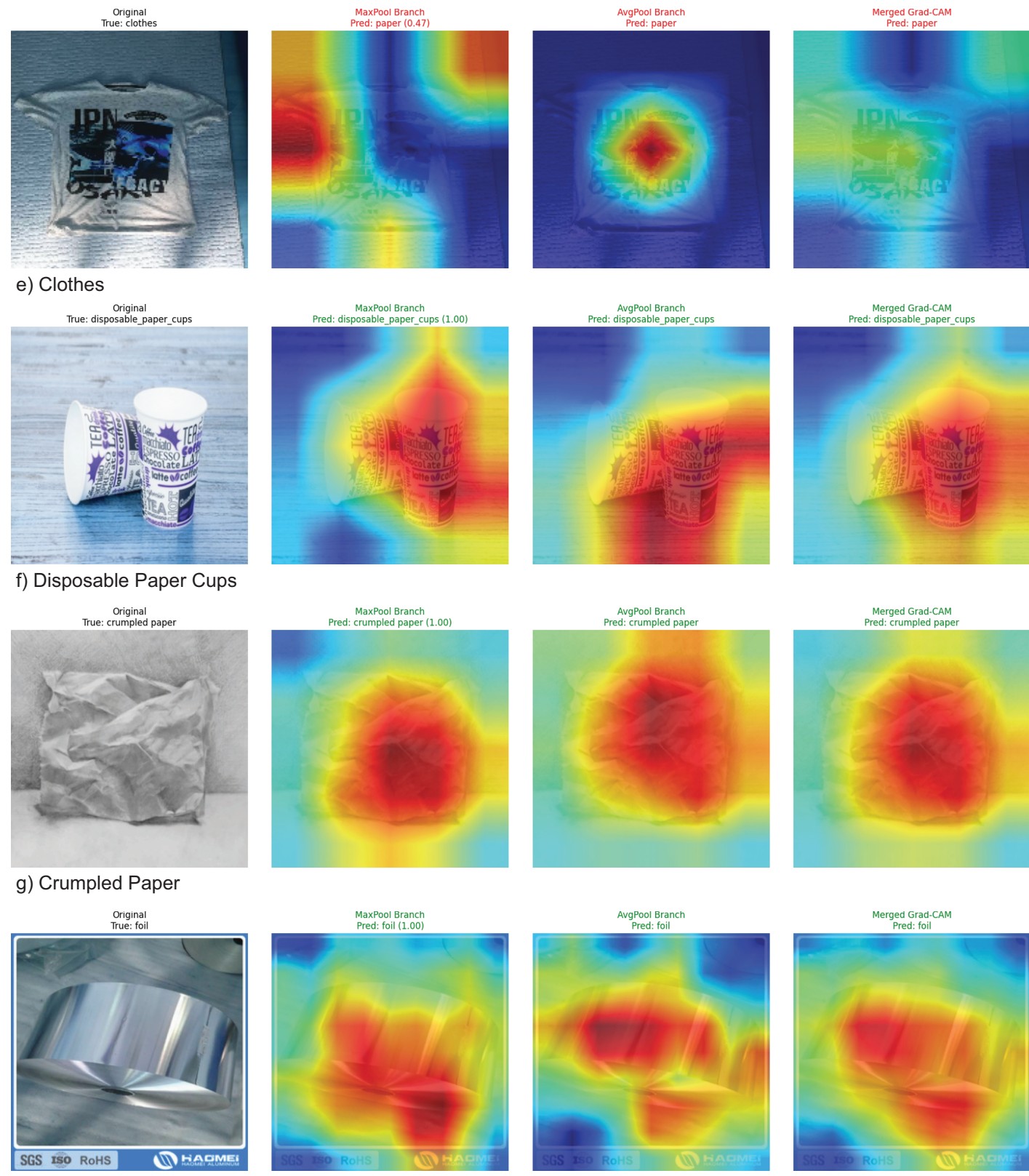

e) Clothes

f) Disposable Paper Cups

g) Crumpled Paper

h) Foil

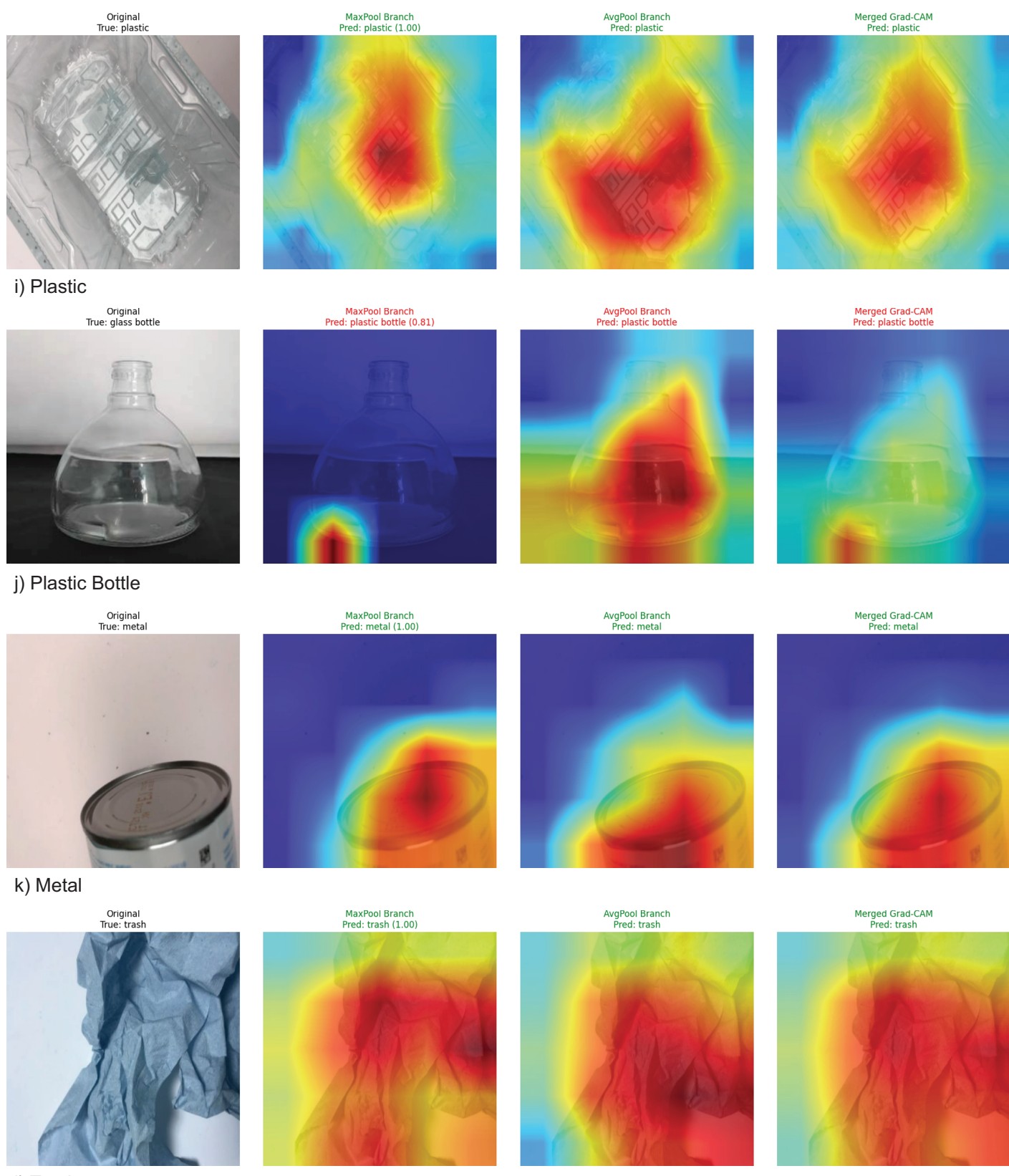

**Fig 20. Explainability of Proposed Model Using Grad-CAM.**

models to identify and classify a wide range of waste into several categories. The decision to implement Flask-based deployment enables the model to operate on a local machine. This approach facilitates offline use without the need for reliable internet access, making it particularly suitable for remote regions.

At the heart of the website's transformative impact on waste management is the remarkable accuracy and compatibility embedded in the H5 file storing the advanced deep learning model, a fusion of DenseNet201 and SE integrated parallel CNN architectures. This meticulously trained model excels in the reliable and precise classification of diverse waste items, recognizing intricate patterns within images for robust classification. As the model performs exceptionally well across a wide range of waste types, it becomes feasible to select one dataset that best aligns with specific regional waste characteristics and utilize the corresponding H5 file for optimal performance. Its compatibility with diverse datasets enables seamless adaptation to a global spectrum of waste types, maximizing practical applications and enhancing inclusivity. This high-accuracy H5 file stands as the linchpin, seamlessly integrating into the website to foster a sustainable and user-driven approach to waste management, marking a significant milestone in the intersection of technology and environmental conservation.

In Fig 21, the user interface of the website unfolds, offering a clear depiction of the intuitive platform. Users are presented with a straightforward design where they can seamlessly upload images of waste items. The interface serves as a gateway to the deep learning models, stored as H5 files, residing within the website's framework. This simplicity ensures a user-friendly experience, encouraging individuals, irrespective of their technical background, to actively participate in the waste classification process.

Moving to Fig 22, a dynamic snapshot captures the moment when a user engages with the platform by uploading a picture of waste. As the image is processed through the intricate layers of the deep learning models, the interface provides real-time feedback, unveiling

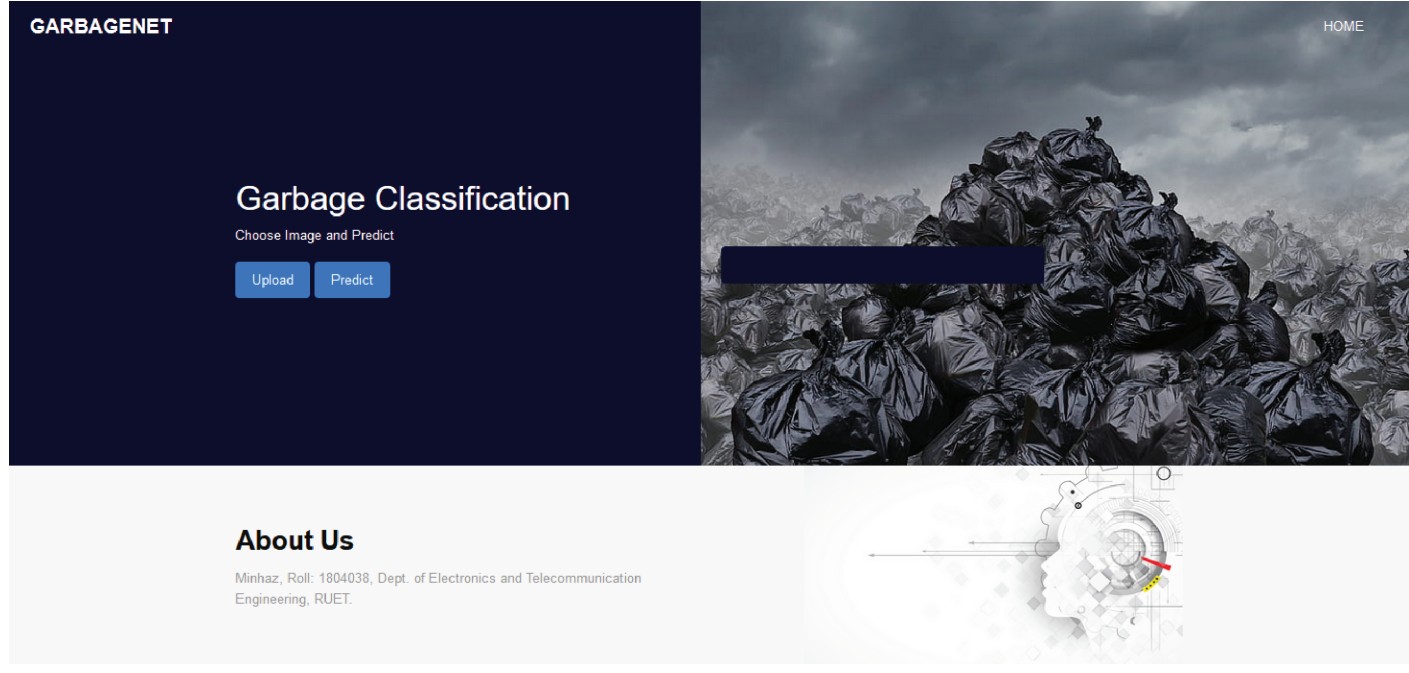

**Fig 21. Interface of the Website.**

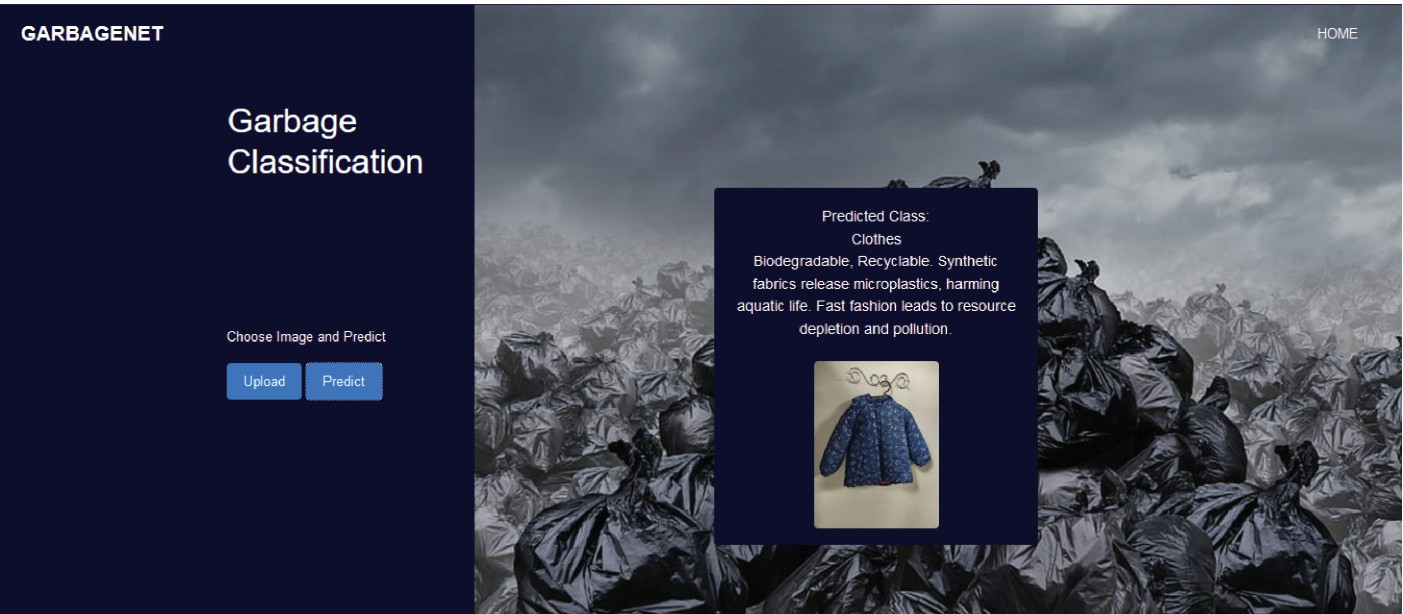

**Fig 22. Uploading an image and predicting via website.**

the predicted class of the waste item. The average end-to-end latency in this case is 0.36 seconds, encompassing image upload time, server processing time, and any response delays. This duration is naturally longer than the model's inference time on the test set. However, for most users, a delay of 0.32 seconds is regarded as very fast and facilitates a smooth user experience. This instantaneous revelation not only enhances the user experience but also reinforces the practicality and efficiency of the website's predictive capabilities. However, if the model is deployed on larger platforms or integrated within broader waste management systems, the latency could fluctuate depending on system load, network conditions, and server capacity. In such scenarios, the response time may increase slightly, although with proper optimization and distributed processing, it can still be kept within acceptable bounds for real-time classification.

## 6.2 Gradio deployment

When it comes to machine learning models, Gradio deployment is a more convenient and effective option than standard website distribution. The Python module removes the complexity involved in web development and provides a high-level, user-friendly interface that streamlines the deployment process. Because of its pre-built components, which allow a variety of input and output types, users may easily design usable interfaces. Gradio's ability to function with a variety of machine learning frameworks without requiring re-implementation is made possible by its model agnosticism. The complexity of traditional website deployment stands in stark contrast to the ease of instantaneous deployment with a single command.

Furthermore, Gradio's ability to handle real-time model updates simplifies iterative development and guarantees that deployed applications remain up-to-date. The deployment experience is further improved by the vibrant community, which offers tools for problem-solving and ongoing development. Essentially, Gradio's features work together to make it easier to

use and more accessible than other website deployment techniques, which is why machine learning applications are better deployed with Gradio.

As can be seen in the graphic representation shown in Fig 23, this deployment allows the user to input an image and have the system automatically generate a class label for it. This flawless procedure highlights the Gradio deployment's efficacy by showcasing its capacity to quickly classify and deliver valuable information based on the supplied image. The deployment's intuitiveness is visually emphasized, highlighting how simple it is for users to retrieve class labels using the interface that is provided.

## 7 Threats to validity

This proposed waste classification model demonstrates exceptional performance, but it does raise some important concerns and limitations. Firstly, the model heavily relies on publicly sourced data, which, although diverse, introduces the possibility of biases and inconsistencies in data quality. Even though certain actions were taken to mitigate this issue throughout the process, some concerns still remain, casting doubts on its practical applicability in real-world waste management scenarios. To address this issue, future efforts could focus on collecting a more diverse and field-specific dataset, as well as real-time data from waste management facilities, grounding the model in the context of practical waste management settings.

Furthermore, the model's ability to generalize to unseen waste types or novel categories remains a concern. This is critical because real-world waste streams evolve continuously, with changes in waste composition—such as new packaging designs, material substitutions, and shifting consumer habits—introducing unfamiliar items. To address this, future work could explore zero-shot learning, domain adaptation, and continual learning strategies. Approaches like incremental retraining and periodic updates with new data can further help sustain model performance over time.

Another notable consideration is the model's relatively high training time and large model size. While it achieves remarkable results, the extended training duration for large datasets

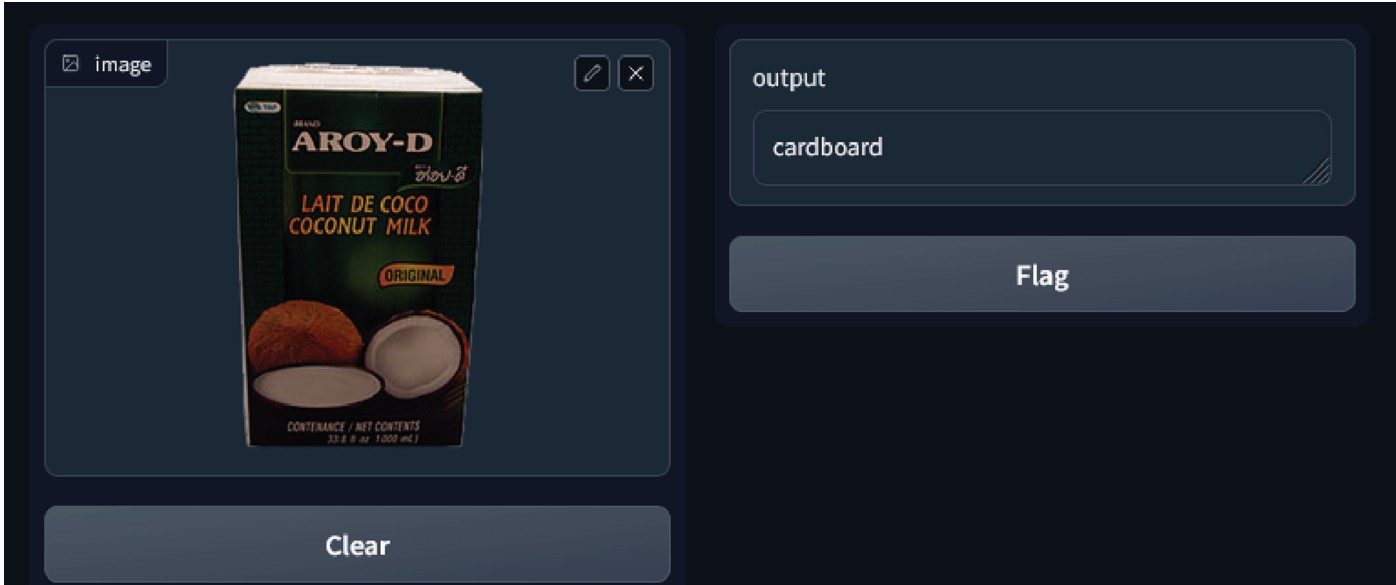

**Fig 23. Gradio Deployment.**

may present challenges in resource-constrained settings, making deployment difficult due to its complexity. To address this, exploring less complex models as alternatives or applying model pruning techniques can help reduce complexity—aiming to retain high accuracy while lowering computational demands and training time.

Additionally, while the successful deployment of the waste classification model on a website represents a significant milestone, the current hosting on a local server offers an opportunity for broader expansion. Future developments could involve extending the deployment to reach a wider audience, potentially integrating the model into a comprehensive waste management system. Large-scale deployments could enable real-time waste classification and optimize waste management across diverse regions. Exploring possibilities, such as deploying the model on devices like the Raspberry Pi, opens avenues for decentralized and on-site waste classification. This not only enhances accessibility but also aligns with the growing trend of edge computing, allowing for more efficient and real-time decision-making in waste management processes.

## 8 Conclusion and future research

In this study, we have developed a sophisticated waste classification model using a modified DenseNet201 architecture complemented by two parallel custom Convolutional Neural Networks (CNN). Each parallel block incorporates a squeeze-and-excitation mechanism, enhancing the model's ability to capture meaningful features from waste images. By concatenating the outputs of these parallel blocks, we have created comprehensive feature maps that contribute to superior classification performance. Our model's utilization of both max pooling and average pooling in parallel blocks further enhances its robustness and adaptability to diverse datasets. Through rigorous experimentation across four distinct waste classification datasets, along with evaluations on three additional datasets, we have consistently achieved outstanding classification accuracy, surpassing existing benchmarks in the field.

The significance of this work lies not only in its exceptional performance but also in its potential to revolutionize waste classification practices. By leveraging state-of-the-art deep learning techniques, we have demonstrated the effectiveness of our model in accurately categorizing waste materials, thereby contributing to the optimization of waste management systems. Additionally, an overview of the model's energy consumption and time complexity has been provided, and Grad-CAM visualization has been incorporated to enhance interpretability by highlighting the critical regions influencing classification decisions. Furthermore, the integration of our model into a Flask-based website enables real-time waste classification, offering practical solutions for waste management practitioners. This accessibility underscores the applicability and scalability of our approach, paving the way for broader adoption and implementation in various waste management contexts.

Upon concluding this study, it is recommended that future investigations concentrate on broadening the diversity of datasets to incorporate a more comprehensive array of waste types and characteristics. To tackle the challenge of generalizing to unseen waste types, future work could explore strategies such as zero-shot learning or domain adaptation. Addressing the challenge of extended training times is paramount for enhancing scalability. Additionally, the current foundation laid by local server hosting paves the way for broader expansion, including the deployment of the model on decentralized devices like the Raspberry Pi. These forthcoming endeavors hold the promise of pushing the boundaries in waste categorization, thereby advancing the capabilities of sustainable waste management practices.

## Author contributions

**Conceptualization:** Md. Minhazul Islam, S. M. Mahedy Hasan, Md. Rakib Hossain, Md. Palash Uddin, Md. Al Mamun.

**Data curation:** Md. Minhazul Islam, S. M. Mahedy Hasan, Md. Rakib Hossain, Md. Palash Uddin.

**Formal analysis:** Md. Rakib Hossain, Md. Al Mamun.

**Investigation:** Md. Palash Uddin, Md. Al Mamun.

**Methodology:** Md. Minhazul Islam, S. M. Mahedy Hasan, Md. Rakib Hossain, Md. Palash Uddin, Md. Al Mamun.

**Resources:** Md. Minhazul Islam.

**Software:** Md. Minhazul Islam, S. M. Mahedy Hasan, Md. Rakib Hossain.

**Supervision:** S. M. Mahedy Hasan, Md. Palash Uddin, Md. Al Mamun.

**Validation:** Md. Minhazul Islam, Md. Rakib Hossain.

**Visualization:** Md. Minhazul Islam, Md. Palash Uddin.

**Writing – original draft:** Md. Minhazul Islam, S. M. Mahedy Hasan, Md. Rakib Hossain.

**Writing – review & editing:** Md. Palash Uddin, Md. Al Mamun.

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
