## [Decision Letter · Decision Letter 0]

9 Aug 2024

PONE-D-24-20413Towards sustainable solutions: effective waste classification framework via enhanced deep convolutional neural networksPLOS ONE

Dear Dr. Uddin,

Thank you for submitting your manuscript to PLOS ONE. After careful consideration, we feel that it has merit but does not fully meet PLOS ONE’s publication criteria as it currently stands. Therefore, we invite you to submit a revised version of the manuscript that addresses the points raised during the review process.

**ACADEMIC EDITOR: Major Revision**

We look forward to receiving your revised manuscript.

Kind regards,

Jyotir Moy Chatterjee

Academic Editor

PLOS ONE

Journal Requirements:

"NO authors have competing interests"

Additional Editor Comments:

The following limitations needs to be addressed:

1. Despite using four publicly available datasets, the study may still lack sufficient diversity to fully represent global waste variations, especially in regions with unique waste profiles.

2. The quality of images in the datasets might vary, which could impact the model’s performance, especially in real-world scenarios where image quality may be inconsistent.

3. While the model was successfully deployed in a web-based system, its scalability to larger, more complex systems or across different geographies remains untested.

4. The model’s performance in diverse environmental conditions (e.g., different lighting or weather conditions) was not explicitly tested, which could affect real-world accuracy.

5. The evaluation was conducted on publicly available datasets rather than real-time data from waste management facilities, potentially limiting the model's real-world applicability.

6. The DenseNet201 architecture and the parallel CNN branches might require significant computational resources, which could be a barrier for deployment in resource-constrained environments.

7. The integration of DenseNet201 with SE mechanisms and parallel CNNs adds complexity, which might make the model more difficult to interpret and deploy.

8. The study does not address potential latency issues in real-time waste classification, which could be critical for practical applications.

9. The study does not explore the potential for deploying the model on edge devices, which could be important for on-site waste management in remote or under-resourced areas.

10. The model might overfit to the specific datasets used for training, reducing its generalization capability to unseen data.

11. If the datasets used are biased towards certain types of waste, the model may not perform well on underrepresented waste types.

12. The study does not thoroughly investigate the model’s robustness to adversarial attacks or noisy data, which are common in real-world scenarios.

13. The energy consumption of the model, especially in large-scale deployments, is not addressed, which could be a significant factor in sustainable waste management.

14. While the study introduces a novel approach, it may not include comprehensive comparisons with a wide range of existing waste classification methods.

15. The long-term stability and performance of the model over time, particularly as waste compositions change, have not been explored.

16. The cost-effectiveness of implementing this advanced model in various waste management contexts has not been analyzed.

17. The complexity of the model may require specialized training for users, which could be a barrier to widespread adoption.

18. The model may not be easily adaptable to specific local waste management needs, which could limit its effectiveness in diverse settings.

19. The deployment of AI in waste management could raise ethical and privacy concerns, especially if data from waste is linked to individuals or households.

20. The web-based deployment of the model assumes reliable internet access, which may not be available in all areas, particularly in remote or developing regions.

Reviewers' comments:

Reviewer's Responses to Questions

**Comments to the Author**

1. Is the manuscript technically sound, and do the data support the conclusions?

Reviewer #1: Yes

2. Has the statistical analysis been performed appropriately and rigorously? 

Reviewer #1: No

3. Have the authors made all data underlying the findings in their manuscript fully available?

Reviewer #1: Yes

4. Is the manuscript presented in an intelligible fashion and written in standard English?

Reviewer #1: Yes

5. Review Comments to the Author

Reviewer #1: The abstract needs modification. The references are to be cited properly. The research gap and limitations are to be included. A tree diagram may be added. The figure title and legends are missing. MCC, Kappa and GDR may be included. Time complexity may be analyzed. A comparison with ML classifier will be included.

6. PLOS authors have the option to publish the peer review history of their article (what does this mean?). If published, this will include your full peer review and any attached files.

Reviewer #1: **Yes: **Harikumar Rajaguru

---

## [Author Response · Author response to Decision Letter 1]

28 Aug 2024

We would like to express our heartfelt gratitude to the academic editor and the reviewer for their valuable time and insightful feedback on our manuscript. We have carefully considered all the comments provided by the academic editor and the reviewer and have made the necessary revisions to the manuscript. The attached "Response to Reviewers" document details how we have responded to the reviewers' concerns. For the ease of the editor and reviewer, all changes in the revised manuscript are highlighted in red.

---

## [Decision Letter · Decision Letter 1]

27 Aug 2024

PONE-D-24-20413R1Towards sustainable solutions: effective waste classification framework via enhanced deep convolutional neural networksPLOS ONE

Dear Dr. Uddin,

Thank you for submitting your manuscript to PLOS ONE. After careful consideration, we feel that it has merit but does not fully meet PLOS ONE’s publication criteria as it currently stands. Therefore, we invite you to submit a revised version of the manuscript that addresses the points raised during the review process.

**ACADEMIC EDITOR: **The editor comments on the paper are not properly addressed by the authors.

We look forward to receiving your revised manuscript.

Kind regards,

Jyotir Moy Chatterjee

Academic Editor

PLOS ONE

Additional Editor Comments:

The editor comments on the paper are not properly addressed by the authors.

Reviewers' comments:

Reviewer's Responses to Questions

**Comments to the Author**

1. If the authors have adequately addressed your comments raised in a previous round of review and you feel that this manuscript is now acceptable for publication, you may indicate that here to bypass the “Comments to the Author” section, enter your conflict of interest statement in the “Confidential to Editor” section, and submit your "Accept" recommendation.

Reviewer #1: All comments have been addressed

2. Is the manuscript technically sound, and do the data support the conclusions?

Reviewer #1: Yes

3. Has the statistical analysis been performed appropriately and rigorously? 

Reviewer #1: N/A

4. Have the authors made all data underlying the findings in their manuscript fully available?

Reviewer #1: Yes

5. Is the manuscript presented in an intelligible fashion and written in standard English?

Reviewer #1: Yes

6. Review Comments to the Author

Reviewer #1: All the corrections are included in the paper. The paper is well modified with all corrective measures and concerns raised by the reviewers. The paper is improved a lot in this revised form.

7. PLOS authors have the option to publish the peer review history of their article (what does this mean?). If published, this will include your full peer review and any attached files.

Reviewer #1: No

---

## [Author Response · Author response to Decision Letter 2]

29 Oct 2024

We sincerely thank the academic editor for their valuable time and insightful feedback on our manuscript. We have carefully considered all the comments provided, recognizing that the previous version did not fully meet expectations. We have made the necessary revisions to address these concerns. The response letter outlines how we have responded to the editor's suggestions. For the ease of the editor, all changes in the revised manuscript are highlighted in red.

---

## [Decision Letter · Decision Letter 2]

9 Mar 2025

PONE-D-24-20413R2Towards sustainable solutions: effective waste classification framework via enhanced deep convolutional neural networksPLOS ONE

Dear Dr. Uddin,

Thank you for submitting your manuscript to PLOS ONE. After careful consideration, we feel that it has merit but does not fully meet PLOS ONE’s publication criteria as it currently stands. Therefore, we invite you to submit a revised version of the manuscript that addresses the points raised during the review process.

We look forward to receiving your revised manuscript.

Kind regards,

Jyotir Moy Chatterjee

Academic Editor

PLOS ONE

Journal Requirements:

Additional Editor Comments:

The work to be revised as follows:

1. The model primarily depends on publicly sourced data, which may introduce biases and inconsistencies in data quality. This limits its applicability to diverse global waste management scenarios, as the datasets may not fully capture variations in waste profiles across different regions

2. The integration of DenseNet201 with squeeze-and-excitation (SE) mechanisms and parallel CNN branches significantly increases computational complexity. This makes it difficult to deploy in resource-constrained environments, such as mobile or embedded systems

3. The study does not thoroughly examine the model’s robustness against adversarial attacks or noisy data. This could impact reliability in real-world scenarios where images may have varying quality due to environmental factors

4. The system has only been deployed on a local server, and its scalability to larger, more complex systems or across different geographies remains untested. Additionally, real-time classification latency and energy consumption in large-scale deployments have not been evaluated

5. The work does not incorporate visualization techniques for explainability, making it difficult to interpret the model’s decision-making process. The lack of explainability tools could hinder trust and adoption by stakeholders who require transparency in AI-based waste classification

Reviewers' comments:

Reviewer's Responses to Questions

**Comments to the Author**

1. If the authors have adequately addressed your comments raised in a previous round of review and you feel that this manuscript is now acceptable for publication, you may indicate that here to bypass the “Comments to the Author” section, enter your conflict of interest statement in the “Confidential to Editor” section, and submit your "Accept" recommendation.

Reviewer #2: (No Response)

2. Is the manuscript technically sound, and do the data support the conclusions?

Reviewer #2: Yes

3. Has the statistical analysis been performed appropriately and rigorously? 

Reviewer #2: Yes

4. Have the authors made all data underlying the findings in their manuscript fully available?

Reviewer #2: Yes

5. Is the manuscript presented in an intelligible fashion and written in standard English?

Reviewer #2: Yes

6. Review Comments to the Author

Reviewer #2: This paper presents a novel approach to waste classification using deep learning techniques, specifically a customized DenseNet201 architecture with parallel CNN branches and squeeze-and-excitation (SE) attention mechanisms. The work is timely and relevant given the growing global challenges of waste management and the need for automated, accurate classification systems.

Strengths:

Comprehensive evaluation: The authors have evaluated their model across four distinct datasets, which is more thorough than many existing studies that focus on a single dataset. This approach demonstrates the model's robustness and adaptability.

Detailed architectural explanation: The paper provides a clear and detailed explanation of the proposed architecture, including justifications for specific design choices like parallel CNN branches and the SE attention mechanism.

Extensive performance metrics: The authors have used a wide range of metrics beyond just accuracy, providing a more complete picture of the model's performance.

Real-world application: The development of a web-based system for deploying the model demonstrates its practical applicability beyond theoretical exploration.

Thorough comparative analysis: The authors have compared their approach against numerous existing models and provided a comprehensive literature review.

Areas for improvement:

Dataset representation: While four datasets are used, there could be deeper discussion on how well these represent global waste variations, especially in regions with unique waste profiles. Consider addressing whether certain geographic-specific waste types might be underrepresented.

Image quality considerations: The performance on varying image qualities is not extensively discussed. Since real-world applications would encounter images of varying qualities, this deserves more attention.

Computational efficiency: While the empirical time complexity is briefly addressed, a more detailed analysis of computational requirements would strengthen the paper, especially considering deployment scenarios with limited computational resources.

Model interpretability: The proposed architecture is complex, which might make interpretation of its decisions challenging. Some discussion on explainability methods that could complement this system would be valuable.

Long-term performance stability: The paper doesn't address how the model might perform over time as waste compositions change. A brief discussion on potential strategies for model updating would enhance the practical relevance.

Technical suggestions:

The confusion matrices in Figures 5, 9, 13, and 17 provide valuable insights, but further analysis of the specific misclassifications (especially between similar materials like paper and cardboard) would strengthen the discussion.

Consider including a brief ablation study to quantify the specific contributions of the parallel CNN branches and the SE attention mechanism to the overall performance improvement.

The empirical time complexity analysis in Table 10 is useful, but could be expanded to include comparisons with other architectures to provide context for deployment considerations.

The web implementation described in Section 6.1 could benefit from more details on potential latency issues in real-time classification scenarios.

Section 6.3 on threats to validity provides good insights, but could be expanded to address potential issues with model generalization to unseen waste types or novel categories.

7. PLOS authors have the option to publish the peer review history of their article (what does this mean?). If published, this will include your full peer review and any attached files.

Reviewer #2: No

---

## [Author Response · Author response to Decision Letter 3]

21 Apr 2025

We sincerely thank the academic editor and the reviewer for their valuable time and insightful feedback on our previously revised manuscript. We have carefully considered all the comments provided, recognizing that the previous version did not fully meet expectations. We have made the necessary revisions to address these concerns. The attached "Response to Reviewers" document outlines how we have responded to the editor's and reviewer’s suggestions. For the ease of the editor and reviewer, all changes in the revised manuscript are highlighted in red.

---

## [Editor Report · Decision Letter 3]

23 Apr 2025

Towards sustainable solutions: effective waste classification framework via enhanced deep convolutional neural networks

PONE-D-24-20413R3

Dear Dr. Uddin,

We’re pleased to inform you that your manuscript has been judged scientifically suitable for publication and will be formally accepted for publication once it meets all outstanding technical requirements.

Kind regards,

Jyotir Moy Chatterjee

Academic Editor

PLOS ONE
---

## [Editor Report · Acceptance letter]

PONE-D-24-20413R3

PLOS ONE

Dear Dr. Uddin,

I'm pleased to inform you that your manuscript has been deemed suitable for publication in PLOS ONE. Congratulations! Your manuscript is now being handed over to our production team.

Kind regards,

on behalf of

Mr. Jyotir Moy Chatterjee

Academic Editor

PLOS ONE